# Ferroelectric solitons crafted in epitaxial bismuth ferrite superlattices

Vivasha Govinden[1,11], Peiran Tong [2,11], Xiangwei Guo [3,4,5,11], Qi Zhang [1], Sukriti Mantri [6], Mohammad Moein Seyfouri [1,7], Sergei Prokhorenko [6], Yousra Nahas [6], Yongjun Wu[3,5], Laurent Bellaiche [6], Tulai Sun[2,8], He Tian [2,9] ✉, Zijian Hong [3,5] ✉, Nagarajan Valanoor [1] ✉ & Daniel Sando [1,10] ✉

In ferroelectrics, complex interactions among various degrees of freedom enable the condensation of topologically protected polarization textures. Known as ferroelectric solitons, these particle-like structures represent a new class of materials with promise for beyond-CMOS technologies due to their ultrafine size and sensitivity to external stimuli. Such polarization textures have scarcely been demonstrated in multiferroics. Here, we present evidence for ferroelectric solitons in $(BiFeO_3)/(SrTiO_3)$ superlattices. High-resolution piezoresponse force microscopy and Cs-corrected high-angle annular dark-field scanning transmission electron microscopy reveal a zoo of topologies, and polarization displacement mapping of planar specimens reveals center-convergent/divergent topological defects as small as 3 nm. Phase-field simulations verify that some of these structures can be classed as bimerons with a topological charge of ±1, and first-principles-based effective Hamiltonian computations show that the coexistence of such structures can lead to non-integer topological charges, a first observation in a $BiFeO_3$-based system. Our results open new opportunities in multiferroic topotronics.

Ferroelectrics are known to exhibit strong coupling of strain to electric polarization. This phenomenon enables the formation of exotic polarization textures which can display emergent properties useful for information storage and sensing devices. Modern developments in fabrication techniques have enabled flexible tailoring of strain, chemical and electrical boundary conditions to achieve a new paradigm of nanoscale complex polarization textures, including topologically protected non-trivial states[1–3]. When such topologically protected polarization textures condense to ultrafine size within a parent medium, they can be considered particle-like objects known as ferroelectric solitons. An example of such three-dimensional ferroelectric solitons are spherical domains (and their transitional states), which

[1]School of Materials Science and Engineering, University of New South Wales Sydney, Kensington, NSW, Australia. [2]Center of Electron Microscopy, School of Materials Science and Engineering, State Key Laboratory of Silicon Materials, Zhejiang University, Hangzhou, Zhejiang, China. [3]School of Materials Science and Engineering, Zhejiang University, Hangzhou, Zhejiang, China. [4]Institute of Advanced Semiconductors & Zhejiang Provincial Key Laboratory of Power Semiconductor Materials and Devices, Hangzhou Innovation Center, Zhejiang University, Hangzhou, Zhejiang, China. [5]Cyrus Tang Center for Sensor Materials and Applications, State Key Laboratory of Silicon Materials, Zhejiang University, Hangzhou, Zhejiang, China. [6]Physics Department and Institute for Nanoscience and Engineering, University of Arkansas, Fayetteville, AR, USA. [7]Solid State and Elemental Analysis Unit, Mark Wainwright Analytical Center, University of New South Wales, Sydney, NSW, Australia. [8]Center for Electron Microscopy, State Key Laboratory Breeding Base of Green Chemistry Synthesis Technology and College of Chemical Engineering, Zhejiang University of Technology, Hangzhou, Zhejiang, China. [9]School of Physics and Microelectronics, Zhengzhou University, Zhengzhou, Henan, China. [10]School of Physical and Chemical Sciences, University of Canterbury, Christchurch, New Zealand. [11]These authors contributed equally: Vivasha Govinden, Peiran Tong, Xiangwei Guo. ✉e-mail: hetian@zju.edu.cn; hongzijian100@zju.edu.cn; nagarajan@unsw.edu.au; daniel.sando@canterbury.ac.nz

possess homogeneously polarized cores surrounded by a curling polarization forming a curved outer shell. The strong polarization and strain gradients that exist at the unit cell level within these structures result in extremely high local crystalline anisotropy, and even symmetries that are energetically unfavorable in the parent bulk. The implications of this are significant: the polarization curling dramatically raises the internal energy, meaning that i) the size of such topological objects is restricted to the nanoscale (i.e., 1–10 nm), and ii) these objects are highly sensitive to external stimulus. These virtues make ferroelectric solitons prime candidates for low-energy and high-density nanoelectronics[4,5].

Ferroelectric solitons have various forms ranging from electrical bubbles[6,7], to polar bubble skyrmions[8] and more recently—thus far only theoretically predicted – hopfions[9]. Other transitional topologies such as merons[10], bimerons and disclinations[11] have also been observed. Interestingly, spherical ferroelectric topologies were in fact theoretically predicted almost two decades ago[12]. Although flux closure structures were previously studied[13–15], it was the demonstration of stable polarization vortex arrays in lead titanate-strontium titanate superlattices by refs. [16] and [1] that triggered a dramatic surge in efforts[17–19]. Following these reports, it was shown that ferroelectric/dielectric superlattices could be tuned to fabricate skyrmion arrays[8] with emergent chiral[20], local negative permittivity[21] and conduction properties[22]. However, such topologies have also been found in simple ferroelectric sandwich heterostructures[23], revealing that ultimately the delicate interplay between the electrical and mechanical boundary conditions drives their formation[12].

The aforementioned work on these complex non-trivial topologies focused on systems made of pure ferroelectrics[7,8] or type-II multiferroics[24] (where the polarization is the secondary order parameter). Solitons have, however, remained elusive in type-I multiferroics. These latter materials also harbor coexisting ferroelectric and magnetic orders, but here the ferroelectric polarization is the primary order parameter meaning that the local crystalline anisotropy (i.e., strain) energy plays a more dominant role than is the case for their type-II cousins. A major type-I room temperature multiferroic candidate is bismuth ferrite (BiFeO$_3$–BFO), which boasts a plethora of appealing properties including multiferroic, photovoltaic, piezoelectric, domain-wall conduction, and optoelectronic responses[25]. In this context, the observation of spherical and transitional topologies such as skyrmions, polar vortex arrays, merons and electrical bubbles in BFO would undoubtedly have wide-reaching implications, both fundamentally and practically. Whilst engineered epitaxial BFO heterostructures have shown writable vortex cores[26], center convergent and quad-domain structures[27] or self-assembled flux closure arrays[5], the observation of topological solitons is still to be achieved. Demonstration of solitons in multiferroic BFO offers exciting prospects over and above their purely ferroelectric counterparts, including new local spin-related physics, and novel ways to engineer spin-lattice coupling. The coexistence of both polar and spin topologies in a single material would enable functionalities such as ultra-fast electric-field control of magnetism, strain, magnetostriction, etc. Moreover, the room-temperature coupling inherent to these systems makes them promising candidates for new spintronic devices. A natural question thus arises: how can one craft the hitherto evasive solitons in BFO?

To address this, one must consider the requirements for the creation of such a polarization configuration in a polar material[28]. The key challenge is to engineer a ferroelectric on the brink, where continuous polarization rotation is achieved without pushing the system into a state where symmetry breaking Ising domain walls are formed[3].

Here, we report the deterministic stabilization of complex topological phases in epitaxial BFO–SrTiO$_3$ (STO) superlattices fabricated on (001)-oriented LaAlO$_3$ (LAO) substrates by pulsed laser deposition. The superlattices display sharp interfaces as revealed by Cs-corrected high-angle annular dark-field scanning transmission electron

microscopy (HAADF-STEM) and atomic resolution energy dispersive spectroscopy (EDS) mapping. Piezoresponse force microscopy (PFM) reveals a diverse range of non-trivial spherical domains, hinting towards the coexistence of a zoo of ferroelectric solitons. Atomic-scale polarization displacement mapping of planar HAADF-STEM specimens confirms the existence of a range of topological structures, from bubbles to bimerons and possibly polar skyrmions. These latter defects show both center-convergent and center-divergent polarization profiles with lateral sizes down to 3 nm. The experimental results are verified with first-principles-based effective Hamiltonian predictions and phase-field simulations, which find that the origin for the formation of these non-trivial topologies is the competing electrical and mechanical boundary conditions. Critically, the simulations also find that the coexisting topological phases can each individually possess either fractional or integer topological charge. The discovery of such ultrafine topologically protected states in multiferroic BFO unlocks an uncharted platform with new degrees of freedom, i.e., control by both electric and magnetic fields.

## Results and discussion
### Experimental design of ferroelectric solitons

Figure 1a graphically depicts the basis of our approach. In ferroelectric/dielectric/ferroelectric structures or ferroelectric/dielectric superlattices, the incorporation of a dielectric layer (often STO) leads to the accumulation of bound charges at the interfaces, thus increasing the system's free energy. If the polarization is oriented out of plane, it is forced to curl towards the in-plane direction to reduce the system's energy, creating the optimal environment for stabilizing ferroelectric solitons. In PbTiO$_3$ (PTO)-based systems, to enable polarization curling, the aim is to force the (naturally oriented) out-of-plane pointing polarization to tilt towards the film plane. This is achieved by using a substrate that imposes an in-plane tensile strain.

In BFO, on the other hand, we use the opposite principle. This is because bulk BFO crystallizes in rhombohedral symmetry with polarization along <111>. A compressive strain is thus required, to push the polarization vector towards the surface normal. For this reason, we use LAO substrates, which impose an in-plane compressive stress that stabilizes an out-of-plane polarization towards the [001] direction (i.e., the so-called T-phase). This strain effect is counterbalanced by a depolarization field created by the insertion of a STO spacer. The spacer has two key effects—(i) it breaks polarization and structural continuity, and (ii) it provides the mechanical compatibility at the interface to allow the polarization to curl.

Although the principle seems simple, growing ultra-smooth layers of epitaxially strained BFO to several tens of nanometers is not trivial. This is where our unique pulsed laser deposition (PLD) chamber system—with a large substrate to target separation (∼ 10 cm)—comes into play. First, this large distance results in low incident flux, enabling controlled ultra-slow layer-by-layer growth of BFO under compressive strain (See Supplementary Note 1 for reflection high energy electron diffraction (RHEED) evidence). Second, the low flux at high temperatures can achieve self-regulated growth of tetragonal like (T-like) BFO to thicknesses up to 60 nm with no mixed phase formation[29]. This ability to fabricate superlattices wherein the long-range in-plane compressive strain can be sustained (i.e., canting the polarization towards the [001] direction of BFO)—somewhat surprisingly as we will see later—is a key first step towards the realization of curling polarization textures in this material.

The next step is to identify the optimal superlattice design. This is critical, as even slight deviations from the optimized thickness can induce strain relaxation mechanisms. The thickness of the BFO layer is pivotal: each layer must be thin enough to maintain the imposed macroscopic strain and dipolar coupling at the interface, without misfit dislocation formation. At the same time, it must also allow coupling across the STO spacers without being influenced by intrinsic

size effects (Fig. 1a). By carrying out a detailed study on the influence of individual layer thickness (Supplementary Note 2) we found the optimal configuration for creating topological textures to be $(BFO_7/STO_4)_{10}$.

Structural and chemical characterization of this optimized system is summarized in Fig. 1a. A representative cross-sectional STEM image within a $(BFO_7/STO_4)_{10}$ heterostructure and its corresponding atomic resolution EDS map (see below) show atomically and chemically sharp heterointerfaces with no interdiffusion (for comprehensive HAADF-STEM and EDS analysis of the complete structure see Supplementary Note 3). Since HAADF is sensitive to the atomic number, the individual BFO and STO layers are seen as alternate bright and dark contrast with thicknesses of 7 and 4 unit-cells (u.c.), respectively, consistent with the intended superlattice design. The well-aligned atomic columns demonstrate high-quality epitaxy without defects. EDS maps reveal distinct Ti (cyan), Fe (green), Sr (yellow), and Bi (red) atomic positions with no interdiffusion, thereby confirming perfect coherent stacking of the BFO and STO layers.

Figure 1b presents symmetrical XRD reciprocal space mapping of a $(BFO_7/STO_4)_{10}$ superlattice near the 002 reflection of film and substrate, revealing various peaks along the out-of-plane ($Q_z$) direction. In addition to the 002 LAO substrate peak, the next brightest spot corresponds to the out-of-plane periodicity of the superlattice with an average out-of-plane parameter of $c = 4.01 \pm 0.01$ Å. Simulations of the diffraction pattern using a custom-made MATLAB program[30] show that the data are consistent with out-of-plane lattice parameters for BFO (STO) of 4.07 Å (3.91 Å) (Supplementary Note 4). The sharp interfaces between the BFO and STO layers lead to additional superlattice reflections such as the $SL_{-1}$, corresponding to the 11 u.c. repeat length. A pertinent feature of this dataset is the horizontal breadth of the main film peak, which could arise from increased local mosaicity, defects, or strain gradients. Since we do not observe chemical defects or dislocations from STEM, we rule out the first two possible effects, implying that significant strain gradients exist within the superlattice structure. The origins of these strain gradients will become clear in Fig. 2. Finally, a peak with narrow horizontal ($Q_x$) breadth is detected at

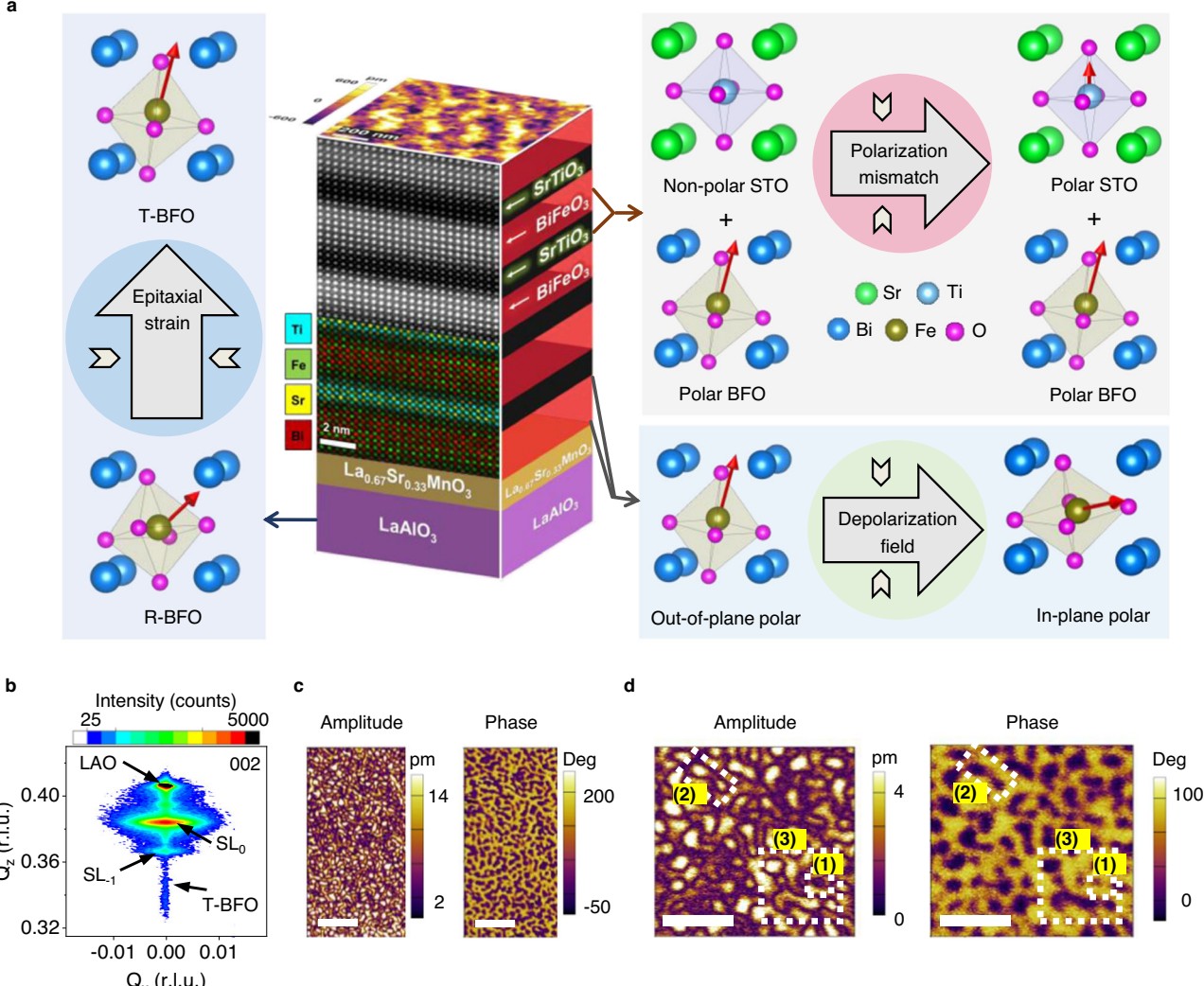

**Fig. 1 | Design of BFO/STO superlattice system and observation of solitons in $(BFO_7/STO_4)_{10}$. a** Schematics of the superlattice with superimposed atomic resolution cross-sectional HAADF-STEM image and atomic EDS-mapping of Ti, Fe, Sr, and Bi showing sharp interfaces. (left) The LAO substrate induces a compressive epitaxial strain favouring a tetragonal-like phase. (right) The introduction of STO has two effects: (i) a polarization mismatch is introduced at the BFO/STO interface causing STO to become polar, and (ii) the polarization discontinuity enhances the depolarization field causing the BFO polarization to curl. **b** Symmetric X-ray diffraction reciprocal space map near the 002 reflection, showing superlattice peaks. **c** PFM amplitude and phase images, revealing complex non-trivial topologies. Scale bars: 200 nm. **d** Higher magnification PFM amplitude and phase images depicting topological structures such as (1) skyrmion, (2) bimeron, and (3) disclination. Scale bars: 100 nm.

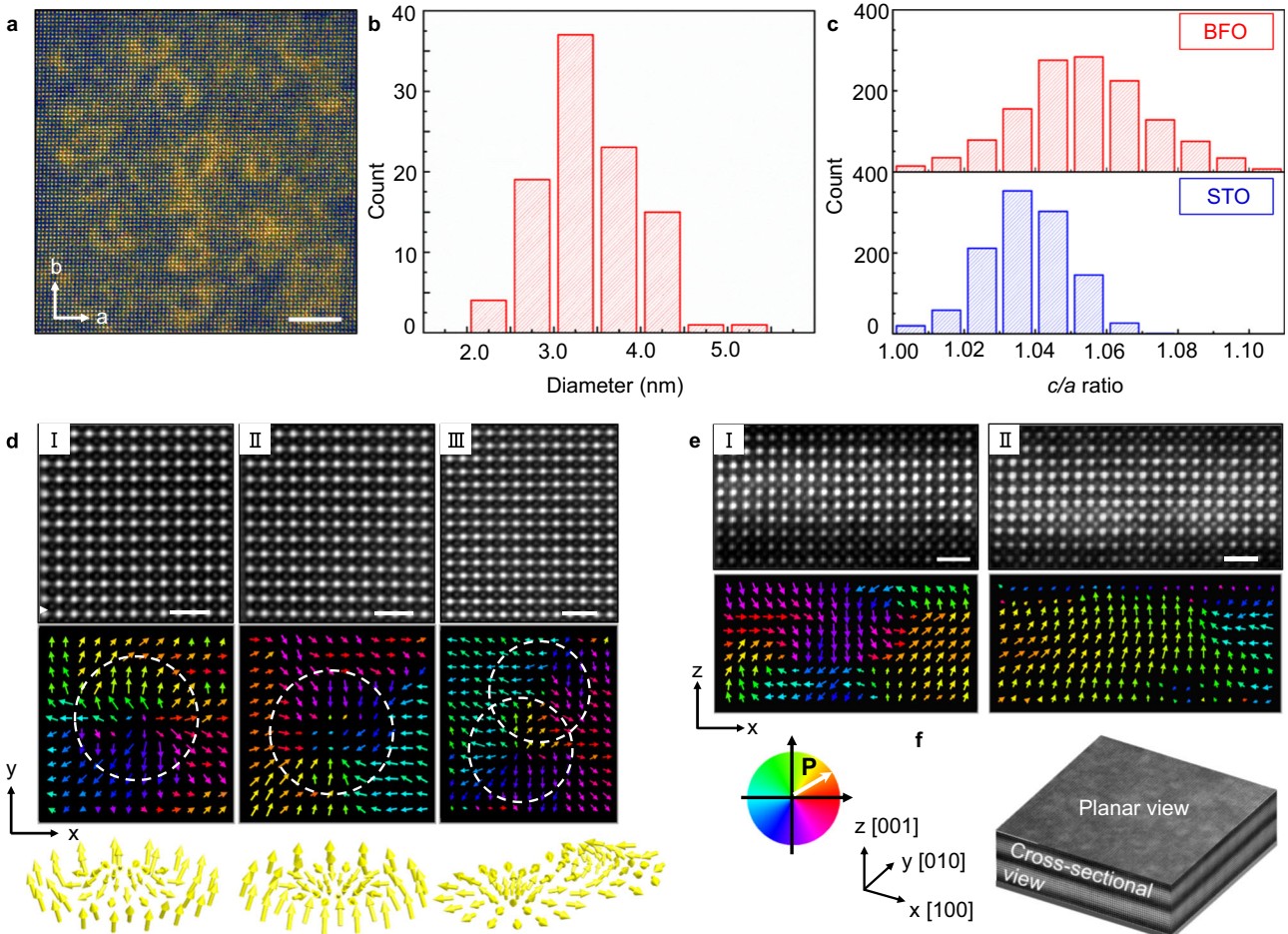

**Fig. 2 | Observation of ferroelectric solitons in BFO/STO superlattices. a** Low-magnification planar-view HAADF-STEM image of the $(BFO_7/STO_4)_{10}$ superlattice. Scale bar: 5 nm. **b** Histogram of size distribution of circular features. **c** The statistics of $c/a$ ratios of STO and BFO extracted from cross-sectional HAADF-STEM images. **d** Enlarged planar-view STEM-HAADF image and corresponding polar vectors, showing (I) center-divergent, (II) center-convergent and (III) bimeron polar textures. Scale bar: 1 nm. **e** Enlarged cross-sectional STEM-HAADF images and corresponding polar vectors, showing (I) an anti-parallel (up-down) polarization, and (II) a trapezoidal shape with convergent polar configuration. Scale bar: 1 nm. **f** Schematics of a planar-view HAADF-STEM image (30 nm × 30 nm) overlaid with a cross-sectional HAADF-STEM image, displaying an overview of the $(BFO_7/STO_4)_{10}$ superlattice.

lower $Q_z$ values, which is indexed as T-like BFO, likely stabilized in the layers closer to the substrate. The average in-plane ($a$) lattice parameter of the $(BFO_7/STO_4)_{10}$ is 3.90 ± 0.01 Å, close to the value of bulk STO (Supplementary Note 4). Thus, the $c/a$ ratio of the entire stack is ~1.04. Later we will compare this value with the results from local measurements of the lattice parameters using STEM. We will show that the local strain mapping reveals that the BFO layers are neither pure T-like nor pure R-like, but rather a mixture of strain and polar states. This multi-strain state is key to relaxing the excess depolarization and elastic energies in this superlattice system.

The critical role played by the LAO substrate now becomes apparent. First, the imposed in-plane compressive strain favors the BFO layer's polarization to be aligned predominantly out of plane. But at the same time, the STO spacer interrupts the polar continuity, inducing strong polarization gradients near the interfaces to minimize the overall increased electrostatic energy costs. These combined effects drive BFO into a state of absolute polar frustration: it can have neither long range out-of-plane polarization, which is prohibited by the cost of depolarization, nor full in-plane tilting, this being restricted by the underlying substrate.

Next, we discuss the non-trivial ferroelectric topologies, imaged using high-resolution dual amplitude resonance tracking (DART) PFM. The amplitude and phase PFM images (Fig. 1c) demonstrate intricate nanoscale domain configurations. Interestingly, we observe

topological features with remarkable semblance to those found by Milde et al. for magnetic skyrmions in $Fe_{1-x}Co_xSi$[31]. A magnified scan (Fig. 1d) reveals the occurrence of sub-20 nm nanoscale features, labeled (1) and outlined by white dashed rectangles. These polarization textures show blurry amplitude contrast and a faint phase reversal at the domain wall. Previously, such features were ascribed to either bubble domains[7] or skyrmions[8,32]. Examples of vector PFM analysis of single center-divergent and center-convergent solitons are presented in Supplementary Note 5. Furthermore, transitional topological defects such as bimerons and disclinations[33] (depicted as (2) and (3) respectively in Fig. 1d) are also identified.

## Atomic scale structural characterization

The resolution limitations of PFM hinder our ability to image any topological feature smaller than ~10 nm[7,34], meaning that this technique alone cannot discriminate between bubbles and skyrmions. The main feature distinguishing polar skyrmions from bubbles is an additional polarization vortex along the circumferential axis, not detectable by PFM. Hence, we next turned to the more precise atomic scale characterization technique of Cs-STEM.

Figure 2a shows a planar-sectional Cs-STEM image of a $(BFO_7/STO_4)_{10}$ superlattice, showing arrays of circular features. The size distribution (Fig. 2b) shows that the structures have typical sizes of ~3.5 nm. Figure 2c shows the statistics of $c/a$ ratios of BFO and STO,

extracted from a cross-sectional STEM image. The lattice parameters of unit cells were calculated by fitting each atom site with a spherical Gaussian using an algorithm in MATLAB. The application of a displacement vector-mapping algorithm on both the cross-sectional and planar view HAADF-STEM images provides direct visualization of atomic-scale polarization displacement within the superlattices (Fig. 2d,e and Supplementary Note 6). Figure 2f is the schematic of the planar-view HAADF-STEM image (30 nm × 30 nm) overlaid with a cross-sectional HAADF-STEM image, displaying an overview of the $(BFO_7/STO_4)_{10}$ superlattice. The polarization vector map is depicted by the color wheel in Fig. 2f.

The planar view HAADF-STEM image and corresponding vector displacement mapping reveal various topological states, as further detailed in Supplementary Note 6. In Fig. 2d, I and II, we observe both in-plane center-divergent and center-convergent polar textures, which take circular form in the low-magnification STEM images (Fig. 2a).

A bimeron-like structure, comprising both center-divergent and anti-vortex polar textures, is also identified (Fig. 2d, III). This structure appears as the fusion of two bubbles in the STEM image of Fig. 2a. Next, in the cross-sectional HAADF-STEM vector displacement mapping, polar regions with anti-parallel (up vs. down) polarizations are found (Fig. 2e,I), with polarization curling near the BFO/STO interfaces. Finally, in Fig. 2e, II, we show a meron-type region with a core consisting of out-of-plane polarization, and as one moves away from the center, the polarization gradually changes into in-plane directions. In Supplementary Note 6, we present planar and cross-sectional STEM data from several other regions that reveal that the solitons are present across the entire specimen. The observation of these out-of-plane and in-plane polar configurations implies, therefore, that we have three-dimensional polar solitons in our $(BFO_7/STO_4)_{10}$ superlattices.

The solitons detected in our samples are qualitatively similar to the skyrmions observed in $(PTO_{16}/STO_{16})_8$ superlattices[8], but our system shows two important distinctions. First, the objects in our $(BFO_7/STO_4)_{10}$ are *not* completely confined within the BFO layers: their polar order also partially exists in the nominally paraelectric STO spacer. We attribute this effect to the strong electrostatic coupling between the thin paraelectric and ferroelectric layers, which causes dipole rotation in the confined STO (ref. [35]). Second, the lower thickness (7 u.c.) of the BFO ferroelectric layer means that our topological objects have a much smaller characteristic size (~3.5 nm) (Fig. 2b) as compared to those in the PTO/STO system (8 nm)[8]. Moreover, note that Fig. 2c reveals a range of $c/a$ ratios with an average of 1.0540, implying that the BFO layers are neither pure T-like, nor pure R-like[36]. To understand the origin of such values, we carried out detailed local strain geometric phase analysis (GPA) mapping results (full details in Supplementary Note 7). First, the GPA confirms the lack of any dislocation cores across the entire BFO/STO superlattice layers, and we do not observe any obvious signs of chemical disorder. The observed wide range of $c/a$ thus should stem from an alternate relaxation mechanism. Noting that the strain mapping shows the superlattices to be in a state of "strain glass," i.e., no long range strain order but consistent with the size scales of the multiple topological solitons (as seen in Figs. 2d and S6) we propose that the superlattices host a plethora of strain states (and hence ferroelectric phases) due to the coexistence of a multiple ferroelectric polarization patterns.

So far, we have shown through PFM and STEM imaging that a range of topological structures exists in our BFO/STO superlattices. However, there is a discrepancy between the size and type of topological features observed by the two characterization techniques. This could be attributed to various factors. First, it is not surprising that different techniques identify diverse objects, given that we have identified a range of topological structures. Second, we point out that during PFM imaging, the trailing field effect of the tip[14], through the applied slight pressure and electric field, can in fact enlarge the topological state. Additional phase-field simulations also confirmed

this external field effect (See Supplementary Note 8 for phase-field simulation evidence). Third, the removal/thinning of the substrate during the preparation of planar view STEM samples dramatically affects the heterostructure's boundary conditions. Arguably, we consider the third case less likely, since in the PTO/STO system, no difference in skyrmion size was identified between the PFM and STEM imaging techniques. Moreover, in our previous work, we did not observe a difference in bubble size for free-standing vs. constrained samples[37]. We have, conversely, shown that bubbles are extremely sensitive to applied scanning probe microscopy pressure and scanning field[7].

Nevertheless, the STEM images are consistent with the previous PFM data. Both techniques show a range of topologies of varying sizes irrespective of the imaging technique. Secondly, both find that the solitons are connected to each other through a complex dipolar network consisting of bimerons and disclination type defects. The fact that we observe such a wide array of topological structures in the BFO/STO superlattices, whereas in the PTO-STO system only a single type of polar skyrmion is observed[16], is likely related to the increased degrees of freedom in the BFO system (cf. the multitude of structural phases evidenced by GPA analysis in Supplementary Note 7). Since BFO is rhombohedral in bulk (i.e., with three cartesian components of the polarization) while PTO is tetragonal in bulk (i.e., with only one cartesian component of the polarization), the polarization vector in the former case has more freedom to form various types of polar arrangements. Moreover, the fact that the electric dipoles in BFO compete/interact with the oxygen octahedral rotations and the magnetic order parameter (while this does not occur in PTO) likely makes the BFO system richer for topological defects.

## Theoretical insight into topological solitons

To gain further insight into the topological polarization configuration and formation mechanism of the experimentally observed polar states, we performed phase-field simulations (details in Supplementary Note 8 and ref. [38]). Figure 3 shows representative polar structures and topological features of a $(BFO_7/STO_4)_8$ superlattice. Low magnification maps of the in-plane polarization component (i.e., planar-like view) (Fig. 3a) show donut-like circular patterns, similar to the polar skyrmion structures observed in the PTO/STO system[8]. However, detailed polar mapping on a finer length scale (Fig. 3b) reveals that the structure in our system is distinct from those in the PTO/STO system. Interestingly, the in-plane polarization (Fig. 3b) forms a combination of center convergent dipole configuration with anti-vortex like arrangement. This is indeed the characteristic of a bimeron and is consistent with the experimental observation by STEM (Fig. 2d).

The topological nature of these features was further characterized by calculating the Pontryagin density, $P_d = \vec{P} \cdot \left( \frac{\partial \vec{P}}{\partial x} \times \frac{\partial \vec{P}}{\partial y} \right)$ [38]. The bubble-like structure shows a circular distribution of the Pontryagin density (Fig. 3c), and a magnified view for a single bimeron yields a ring-like feature (Fig. 3d). Surface integration of the circle gives a topological charge of −1, confirming that such structures are indeed more like bimerons. Furthermore, both center-divergent and center-convergent type structures are observed in this system (Supplementary Note 8), once again in good agreement with experiment. Further insight is gained by plotting a cross-section view of the simulation (Fig. 3e,f). A 180° domain wall like structure, with alternating positive and negative out-of-plane polarization components (Fig. 3e), is found. The plot of the polarization vector (Fig. 3f) shows the formation of a polar vortex like structure. It is also of note that in the BFO layer, both R-like and T-like regions are identified, in agreement with the experimental observations. In addition, the equilibrium structure in $BFO_n/STO_4$ superlattice heterostructures with varying BFO layer thicknesses was studied by phase-field simulations. It was discovered that when the BFO layer thickness decreases from 28 unit cells, the BFO layer undergoes a topological phase transition from twin domains (for

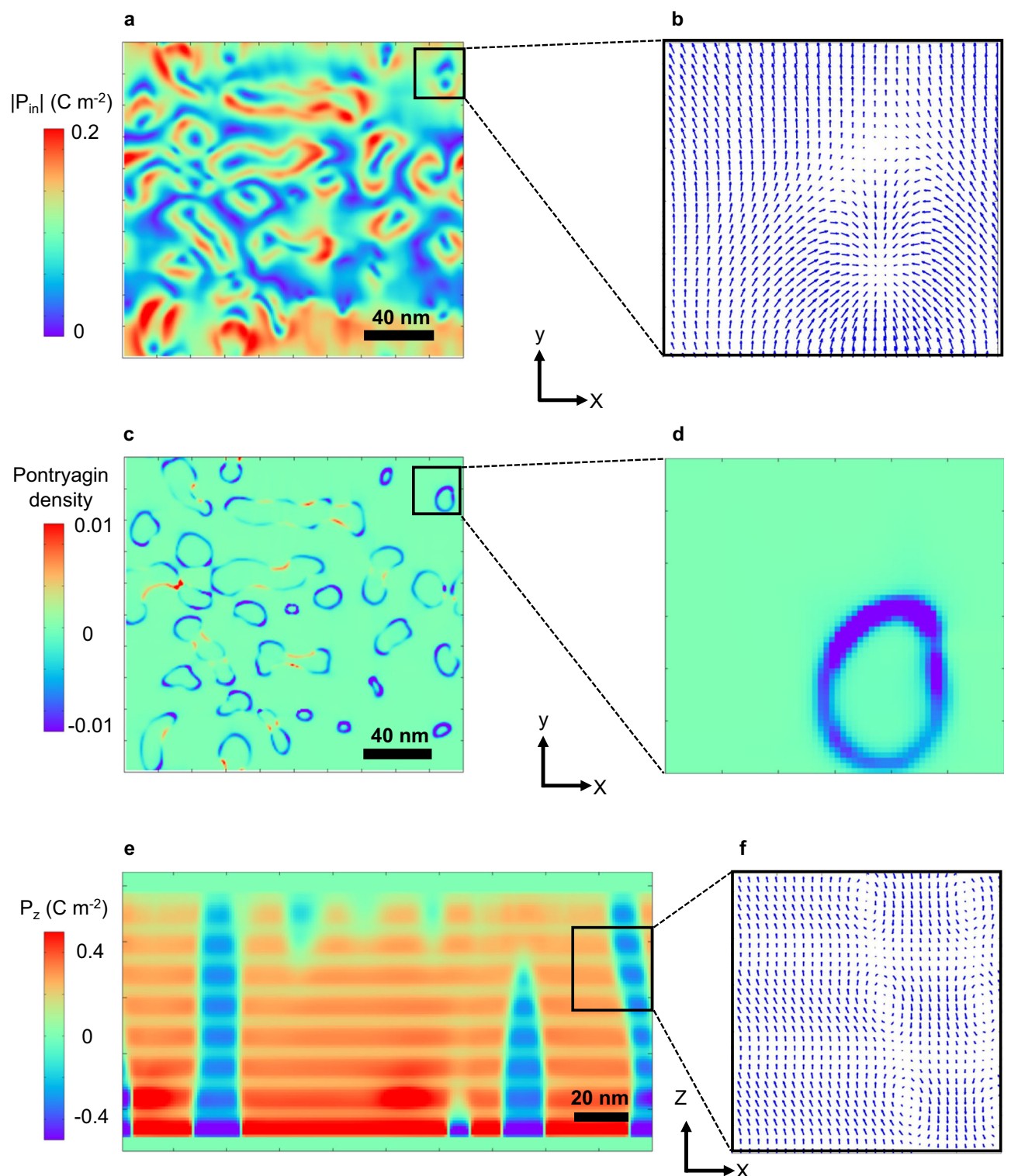

**Fig. 3 | Polar structures and Pontryagin density of (BFO$_7$/STO$_4$)$_8$ superlattices calculated by phase-field simulations. a** Planar view of the in-plane polarization magnitude. **b** Magnified view of the in-plane polar vector plot, showing a bimeron structure. **c** Plot of the Pontryagin density for the same region as shown in (**a**). **d** Magnified view of the Pontryagin density of the bimeron structure. Surface integration of the Pontryagin density for the bimeron gives a topological charge of −1. **e** Cross-section view of the out-of-plane polarization magnitude, showing alternating positive and negative polarization. **f** Magnified view of the out-of-plane polar vector plot, showing a vortex-like structure, with a mixture of R-like and T-like polar regions.

$n > 11$) to solitons ($n$ between 5 and 10) to monodomain ($n < 5$). The main driving force behind this topological phase transition is the competition between bulk and elastic energy densities, as shown in Fig. S13 (Supplementary Note 8).

To further strengthen our predictions of the formation of topological objects under compressive strain, we also used a first-principles-based effective Hamiltonian[39] to study thin film BFO. A 16 u.c. thick (001) oriented BFO film with an initial 109° domain

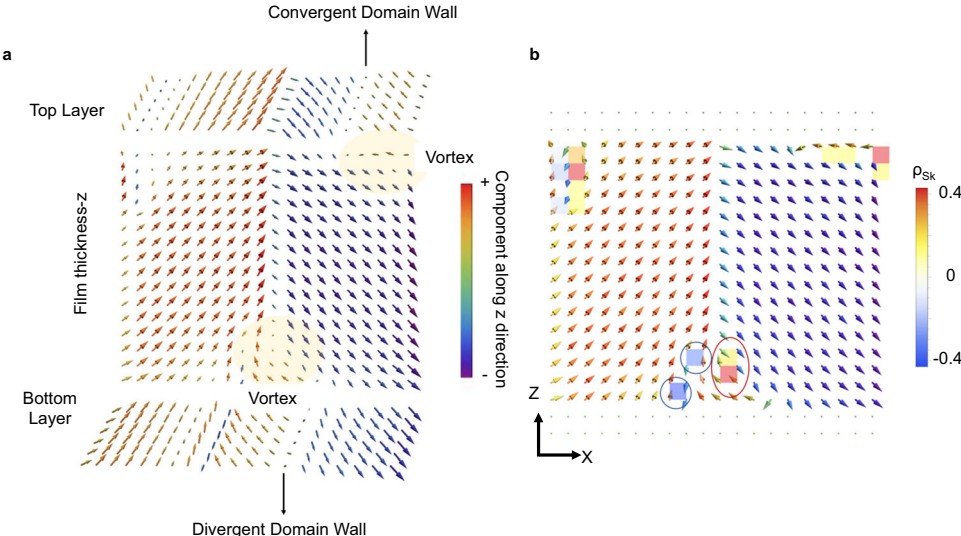

**Fig. 4 | Polar configurations computed by effective Hamiltonian calculations.** **a** Polar structure (arrows) of a BiFeO$_3$ thin film supercell after relaxation from a 109° domain structure under −2.9% epitaxial strain at 10 K. The $x$-$z$ plane is shown along with the top and bottom most layers of the film. **b** Distribution of the Pontryagin's charge density (colored plaquettes) along with the normalized dipoles (arrows) for the polar distribution shown in (**a**).

structure was placed under mechanical boundary conditions of epitaxial compressive strain varying up to −5%, with open-circuit-like electrical boundary conditions. A thickness of 16 unit cells was chosen to better show the vortex features. However, thickness equal to the experiments also showed similar polar mode features (Supplementary Note 9). Multidomain structures are known to be preferred over monodomains under such boundary conditions[12]. The specific multidomain structure of 109° domain type was chosen as it has been the most experimentally observed domain wall in bulk BiFeO$_3$ and possesses the lowest domain wall energy according to first-principles-based calculations[40]. The film was modeled by a supercell periodic in the $x$ and $y$ directions. Monte Carlo simulations were run for 50,000 steps at 10 K for each value of strain. It was observed that the initial 109° domains first changed into a topologically non-trivial structure (Fig. 4a), which had (i) a vortex in the $x$–$z$ plane, and (ii) convergent/divergent polar textures in the $x$–$y$ plane located at the circumference of the vortices.

The features seen in these computed polar structures agree with HAADF-STEM experiments. Namely, effective Hamiltonian simulations reproduce the zig-zag vortex pattern observed in cross-sectional HAADF-STEM images (pattern I in Fig. 2e) while convergent/divergent lines at the BFO interfaces in Fig. 4a support the formation of center convergent/divergent points observed in Fig. 2d. The vortex occurs in a moderate compressive strain region of [−3%, −2%], beyond which it is destroyed. The $c/a$ ratio of BFO for the substrate strains that result in these vortex states is 1.077 and 1.050 at misfit strains of −3% and −2%, respectively, consistent with the $c/a$ ratios observed experimentally.

To analyze the topology of the resultant dipolar structure, we calculated the distribution of the Pontryagin's charge density ($\rho_{Sk} = \mathbf{n} \cdot (\partial_x \mathbf{n} \times \partial_y \mathbf{n})$, where $\mathbf{n}$ denotes normalized electric dipoles) in the $x$-$z$ plane. The distribution of the resulting density (colored plaquettes) and the normalized dipoles (arrows) are shown in Fig. 4b where red and blue circles indicate examples of meron and antimeron lines. Interestingly, these lines carry a *fractional* Skyrmion number. Particularly, the meron indicated by a red circle in Fig. 4b has a Skyrmion number of 0.54, while the anti-merons outlined in blue carry charges of −0.43 and −0.5. Notably, despite the presence of both positively and negatively charged merons, the integral Skyrmion number ($\int (dx\, dy\, \rho_{Sk})$) over the simulation supercell was found to be equal to one.

In summary, we have crafted polar soliton structures in epitaxial multiferroic BFO/STO superlattices. The demonstration of these previously elusive topological states in this material system is anticipated to have far reaching implications on the landscape of topological polar/spin textures in multiferroics. Given that the various Dzyaloshinskii-Moriya interactions—which govern the weak ferromagnetic moment and the spin cycloid in BFO[41]—are driven by strain and local symmetry, one can anticipate that solitons in BFO could reveal enhanced local ferromagnetic moments, modified magnetic transition temperatures, and/or enhanced electrical conductivity. Further, similar to how domain walls in materials such as TbMnO$_3$ can harbor exotic electronic and magnetic states[42], our solitons may indeed constitute a fundamentally different multiferroic phase of BFO. These findings are clearly just the tip of the iceberg; we hope that our results will motivate practitioners and theorists in the field to dig deeper into these superlattice systems. Future work will require elucidating the specific role of the local symmetry changes within the solitons and how it influences the local polarization dynamics, weak ferromagnetic moment, optical behavior, and transport responses- all functionalities that can be used in next generation nanoscale devices.

## Methods

### Epitaxial thin-film fabrication

A range of bismuth ferrite (BiFeO$_3$; BFO)–strontium titanate (SrTiO$_3$; STO) superlattices, i.e., (BFO$_m$/STO$_n$)$_{10}$ films were synthesized on La$_{0.67}$Sr$_{0.33}$MnO$_3$ (LSMO) buffered (001) oriented LaAlO$_3$ (LAO) substrates (Shinkosha, Japan) by pulsed laser deposition (PLD, Neocera, USA) where $m$ and $n$ are the number of unit cells (u.c.) of BFO and STO respectively. LSMO of 5 nm thickness was deposited at 800 °C under 100 mTorr oxygen partial pressure while BFO and STO were deposited at 700 °C under 23 mTorr oxygen partial pressure. The thickness of each individual BFO layer was varied from 4 to 7 u.c. while that of the STO layer was varied from 2 to 4 u.c. Polar solitons were found in (BFO$_7$/STO$_4$)$_{10}$ with optimal thickness of the BFO and STO layers of $m = 7$ and $n = 4$ u.c., respectively. Note that for all samples, there was also a capping BFO layer of the same thickness $m$.

### Structural characterization

Conventional $\theta$−2$\theta$ scans and reciprocal space maps were performed using Cu $K_{\alpha-1}$ radiation in a 9-kW rotating anode Rigaku SmartLab diffractometer. A custom-made MATLAB program[30] was used to

simulate the XRD pattern of the desired heterostructure. This program uses thickness and out-of-plane lattice parameters as input variables. Comparing the experimental data to these simulated patterns allows the accurate determination of the thicknesses $(m, n)$ and out-of-plane $c$ lattice parameter of the individual layers.

## Scanning probe microscopy

The topography and domain patterns of the as-grown state for all films were obtained on a commercial scanning probe microscope (Cypher S, Asylum Research, US) using Pt/Cr coated probes (Multi75GE, Budget-Sensors, Bulgaria) under a force of <100 nN. The force constant and free resonance frequency of the probe cantilever were $3\,N\,m^{-1}$ and 75 kHz, respectively.

## Scanning transmission electron microscopy

Atomic resolution high-angle annular dark field (HAADF) scanning transmission electron microscopy (STEM) was carried out. Planar view samples (observation along the [001] direction) and cross-sectional view samples (observation along the [010] direction) were prepared by focused ion beam (FIB) (FEI Quanta 3D FEG) for observation by transmission electron microscopy. In consideration of the damage caused by ion beam in FIB, we used low voltage and beam current of 2 kV/27 pA. The observations were performed by spherical aberration-corrected electron microscopy on a FEI Titan G2 80–200 ChemiSTEM (30 mrad convergence angle, 0.8 Å spatial resolution), equipped with Super-X energy-dispersive X-ray spectroscopy (EDS) with four windowless silicon-drift detectors. The positions of the atomic columns in the images were confirmed by a mathematical method involving Gaussian Fitting based on MATLAB. Polarization mapping was performed by calculating ion displacements in the HAADF-STEM images. The microscopy data for quantitative analysis were acquired under the condition of the sample drifting less than $1\,Å\,min^{-1}$. Atomic-resolution EDS mapping was performed with an electron beam current of ~100 pA and a dwell time of 10 μs per pixel.

## Phase-field simulations

Phase-field simulations were performed to simulate the equilibrium polar structure for $[(BiFeO_3)_7/(SrTiO_3)_4]_8$ ($(BFO)_7/(STO)_4$ in short) grown on an $LaAlO_3$ substrate. In the film layer, seven unit cells of BFO and 4 unit cells of STO are deposited periodically, which is consistent with the experiment. Two sets of order parameters were used in the present model, that is, the spontaneous polarization vector ($\boldsymbol{P}$), and oxygen octahedral tilt ($\boldsymbol{\theta}$). The kinetics of both order parameters are described by the time-dependent Ginzburg-Landau equation as follows:

$$\frac{\partial \phi}{\partial t} = -M \frac{\delta F}{\delta \phi} \tag{1}$$

where $t$, $M$ and $\phi$ denote the evolution time step, the kinetic coefficient, and the order parameter (either $\boldsymbol{P}$ or $\boldsymbol{\theta}$), respectively. The free energy $F$ has the contributions from the individual energy densities, i.e., the Landau/chemical, elastic, electrostatic, and polar/rotation gradient energy densities:

$$F = \int \left( f_{\text{Landau}} + f_{\text{elastic}} + f_{\text{electric}} + f_{\text{gradient}} \right) dV \tag{2}$$

Detailed expressions of these energy densities, the numerical treatments, the phase-field equations, and the parameters are described in refs. [43],[44].

A series of three-dimensional meshes of $200 \times 200 \times 150$ were used, with a grid spacing of 0.4 nm. Along the $z$ direction, from bottom to top, the thickness of the substrate, thin film and air was set to be 30,

87, and 33 grids, respectively. Periodic boundary conditions were applied in the two in-plane dimensions, while a superposition method was employed along the thickness dimension[45]. A short-circuit electric boundary condition was adopted, where the electric potential at the top of the film and at the film/substrate interface was zero[46]. The elastic boundary condition was chosen to be stress-free on the film top, and zero displacement on the bottom of the substrate sufficiently far away from the film/substrate interface. Random noise was added with a small magnitude ($<0.0001\,\mu C\,cm^{-2}$) to simulate the initial nuclei during film growth.

## Effective Hamiltonian computations

An initial 109° domain structure in BFO was relaxed through Monte Carlo simulations using a first-principles based effective Hamiltonian, whose energy and sub-parts are shown in the following equation (a detailed description can also be found in Ref. [39]):

$$E_{\text{total}} = E_{\text{polar,strain}}(\{u\},\{\eta\},\{v\}) + E_{\text{tilt}}(\{\omega\},\{u\},\{\eta\},\{v\})$$
$$+ E_{\text{mag}}(\{\omega\},\{u\},\{\eta\},\{v\},\{m\}) + E_{depol}(\{u\},\beta) \tag{3}$$

$$E_{\text{polar,strain}} = \sum_i \left[ \kappa_2 u_i^2 + \alpha u_i^4 + \gamma \left( u_{ix}^2 u_{iy}^2 + u_{iy}^2 u_{iz}^2 + u_{ix}^2 u_{iz}^2 \right) \right]$$
$$+ \sum_{ij} \sum_{\alpha\beta} Q_{ij,\alpha\beta} u_{i,\alpha} u_{i,\beta} + \sum_{i \neq j} \sum_{\alpha\beta} J_{ij,\alpha\beta} u_{i,\alpha} u_{i,\beta}$$
$$+ E_{\text{Homogeneous strain}}\{\eta\} + E_{\text{Inhomogeneous strain}}\{v\}$$
$$+ 1/2 \sum_i \sum_{\alpha\beta} B_{l\alpha\beta} \eta_l(i) u_{i\alpha} u_{i\beta} \tag{3.1}$$

$$E_{\text{tilt}} = \sum_i \left[ \kappa_A \omega_i^2 + \alpha_A \omega_i^4 + \gamma_A \left( \omega_{ix}^2 \omega_{iy}^2 + \omega_{iy}^2 \omega_{iz}^2 + \omega_{ix}^2 \omega_{iz}^2 \right) \right]$$
$$+ \sum_{ij} \sum_{\alpha\beta} K_{ij\alpha\beta} \omega_{i\alpha} \omega_{j\beta} + \sum_i \sum_{\alpha} K' \omega_{i,\alpha}^3 \left( \omega_{i+\alpha,\alpha} + \omega_{i-\alpha,\alpha} \right)$$
$$+ \sum_i \sum_{\alpha\beta} C_{l\alpha\beta} \eta_l(i) \omega_{i\alpha} \omega_{i\beta} + \sum_{ij} \sum_{\alpha\beta} D_{ij,\alpha\beta} u_{j,\alpha} \omega_{i,\beta}$$
$$+ \sum_{ij} \sum_{\alpha\beta\gamma\delta} E_{\alpha\beta\gamma\delta} \omega_{i\alpha} \omega_{j\beta} u_{j\gamma} u_{i\delta} \tag{3.2}$$

$$E_{\text{mag}} = \sum_{ij\alpha\gamma} Q_{ij\alpha\gamma} m_{i\alpha} m_{j\gamma} + \sum_{ij\alpha\gamma} D_{ij\alpha\gamma} m_{i\alpha} m_{j\gamma} + \sum_{ij\alpha\gamma\nu\delta} E_{ij\alpha\gamma\nu\delta} m_{i\alpha} m_{j\gamma} u_{i\nu} u_{i\delta}$$
$$+ \sum_{ijl\alpha\gamma\nu\delta} F_{ij\alpha\gamma\nu\delta} m_{i\alpha} m_{j\gamma} \omega_{i\nu} \omega_{i\delta} + \sum_{ijl\alpha\gamma} G_{ijl\alpha\gamma} \eta_l(i) m_{i\alpha} m_{j\gamma}$$
$$+ \sum_{ij} K_{ij} \left( \omega_i - \omega_j \right) \cdot \left( m_i \times m_j \right) \tag{3.3}$$

$$E_{depol} = \beta \sum_i <E_{\text{depolarization}}> \cdot u_i \tag{3.4}$$

Here, $u_i$ represents the polar soft mode centered on the Bi ion in the $i$th unit cell; $\omega_i$ represents the tilt of the oxygen octahedra with its direction being the axis of tilt and magnitude representing the angle of tilt in radians; $\eta$ is the global homogenous strain tensor in the Voigt notation; $v_i$ represents (Fe-centered) variables related to the inhomogeneous strain tensor distorting the unit cell; $m_i$ is the magnetic moment centered at the Fe site with its magnitude fixed to be $4\,\mu_B$ (consistent with first-principles calculations and measurements) and $\beta$ quantifies the capability of the system to screen polarization-induced surface charges[12]. The first term (3.1) is the energy due to local soft modes, homogenous strain tensor, inhomogeneous strains, and their mutual couplings. The second term (3.2) is the energy due to anti-ferrodistortive tilt modes, and their coupling with local soft modes and strains. The

third term (3.3) is the energy due to magnetic moments (exchange and dipolar interactions) and their couplings with local soft modes, strains and anti-ferrodistortive tilt modes. The fourth term (3.4) is the depolarization energy. All the effective Hamiltonian coefficients are calculated from first principles.

In this paper, an open-circuit electrical boundary condition is used, *i.e.*, $\beta = 0$. To incorporate the effect of epitaxial strain, the three in-plane strain components

$$\eta_1 = \eta_2 = \frac{a_{sub} - a_{ref}}{a_{ref}}, \eta_6 = 0 \tag{4}$$

were fixed while all others were allowed to freely relax, where $a_{sub}$ is the in-plane lattice constant of the substrate and $a_{ref}$ is the lattice constant of the cubic bulk BFO calculated via first principles at 0 K. The strain is then calibrated corresponding to the ground state (lowest energy) strain of the epitaxially strained film.

The Pontryagin's charge density was calculated using an in-house developed topological charge calculator based on the Berg and Lüscher discretization approach for computing the $\pi_2$ class indices[47]. For charge calculations, we used the relaxed polar mode output of the supercell to produce the resultant cell-by-cell topological charge mapping in the *x-z* plane.

## Data availability

All data used are available within this manuscript and Supplementary Information. Further information can be acquired from the corresponding authors upon reasonable request.

## Code availability

The phase-field simulation results in this work were obtained using the software package Mu-PRO (www.mupro.co).

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

## Acknowledgements

The research at University of New South Wales (UNSW) is supported by DARPA Grant No. HR0011727183-D18AP00010 (TEE Program), partially supported by the Australian Research Council Centre of Excellence in Future Low-Energy Electronics Technologies (project number CE170100039) and funded by the Australian Government. This work is also supported by the National Natural Science Foundation of China (grant nos. 92166104 and 12125407), the Zhejiang Provincial Natural Science Foundation (LD21E020002), the Joint Funds of the National Natural Science Foundation of China (U21A2067), the National Key Research and Development Program of China (No. 2021YFA1500800). X.G. is supported by the National Natural Science Foundation of China (No. 52202151) and China Postdoctoral Science Foundation (No. 2022M722715). Z.H. also gratefully acknowledges a start-up grant from Zhejiang University. Q.Z. acknowledges the support of a Women in FLEET Fellowship. The research at University of Arkansas is supported by ARO Grant No. W911NF-21-1-0113, the Vannevar Bush Faculty Fellowship (VBFF) Grant No. N00014-20–1-2834 from the Department of Defense, ARO Grant Number and ARO grant number W911NF-21-2-0162 (MURI-ETHOS). We also acknowledge the Arkansas High Performance Computing Center (AHPCC). We thank S. Prosandeev for assistance with effective Hamiltonian calculations. L.B. dedicates this paper to Professor Henry Krakauer, who made pioneering contributions to the understanding of ferroics via the development and use of original numerical techniques.

## Author contributions

V.G. fabricated the samples. V.G., M.S., and Q.Z. performed scanning probe microscopy and analyzed the data. V.G., M.S., and D.S. carried out X-ray diffraction and analysis. P.R.T., T.L.S., and H.T. performed electron microscopy and detailed polarization vector analysis. X.G., Y.W., and Z.H. performed phase-field simulations. S.M., S.P., Y.N., and L.B. performed effective Hamiltonian calculations. V.G., D.S., and N.V wrote the manuscript with input from other authors. N.V. and D.S. directed the research. All authors contributed to the discussion, analysis, and manuscript preparation.

## Competing interests

The authors declare no competing interests.
