## [Peer Review File · Nature Communications]

Ferroelectric Solitons Crafted in Epitaxial Bismuth Ferrite SuperlatticesEditorial Note: Parts of this Peer Review File have been redacted as indicated to maintain the confidentiality of unpublished data and to remove third-party material where no permission to publish could be obtained.

REVIEWER COMMENTS

Reviewer #1 (Remarks to the Author):

This work done by V. Govinden et al. reported the observation of particular ferroelectric topological domains, see ferroelectric solitons, in BFO-STO superlattice sample prepared by PLD. Detailed PFM and TEM results were acquired, and the manuscript was well organized. As the author said, there is an obvious inconsistency between PFM results and TEM results in this paper. Although the author has listed several possibilities, the contradiction between the results cannot be ruled out. The correlation between TEM data and PFM data is worth rethinking, otherwise it is difficult to support the existing conclusions. My main comments are as follows:

1. For this kind of superlattice specimens, plane-view sample was selected to visualize the in-plane polarization distribution. It is well known that TEM samples are usually tens of nanometers thick. Therefore, the electron beam will encounter multiple periodic alternating STO/BFO layers. How to interpret the collected HAADF images is worth discussing. Authors are advised to add STEM simulations to enhance the interpretation of the HAADF images, the existing polarization mapping in Fig.2 is not strong evidence.

2. I can not agree with the authors claimed that the PFM tip enlarges the topological state. The size of a single topological domain is less than 3nm (line 102), which is beyond the detection capability of PFM, there is no so-called "amplification" effect here. At the very least, I don't think it can be used as a reason to explain the gap between PFM and TEM data.

Some suggestions are as follows:

1.(line 87) It has been reported that the polar vortex has been observed in BFO multi-iron films, which is similar to the ferroelectric soliton mentioned by the authors.

(<https://arxiv.org/pdf/1810.12895v1.pdf>) Can authors relate the topology to the macroscopic physical properties?

2.(line 182) In Fig. 1d and e, I suggest the author to calibrate the polarization orientation via conventional vector PFM method, and determine the polarization distribution of the skyrmions structure.

3. In line 191, it is no evidence to claim the resolution of PFM cannot resolve domain structures less than 20 nm. The tip radius is less than 10 nm (Rocky Mountain 12PT400B), and it is not a problem to resolve domain structures smaller than 20 nm.

4.(line 216) The authors claimed that 3D ferroelectric solitons, such as spherical domains, have been observed in BFO/STO superlattice films. Can the 3D structure of ferroelectric solitons be reconstructed according to the in-plane and out-of-plane polarization profiles in Fig.2? And provide more evidences, such as 4D-STEM?

5.(line 223) The polar mapping results based on HAADF-STEM show that the size of ferroelectric soliton is only 3 nm, while the size of ferroelectric soliton shown by PFM in Fig. 1d and e is over 20nm. This difference makes me doubt whether PFM and TEM observe the same object. Authors are advised to provide HAADF-STEM results, which show multiple soliton configurations or soliton array structure, maybe corresponding 3D-RSM results with apparent periodic structure in plane, similar to Reference 15?

6. In Fig.2, BFO/STO interface is not sharp, including atomic EDS results in Fig. S3. Interface diffusion? or film surface not smooth? From polar mapping, the polarization in STO layer is larger than that in BFO. Can the authors provide more polar mapping results with large scale?

7. In Fig.S2, Can the corresponding topography be provided to exclude the influence of

topography on PFM amplitude?

Reviewer #2 (Remarks to the Author):

Over the past two decades, polar topological structures in ferroelectrics have attracted intensive attention for their potential as the building blocks in developing nanoelectronics. In this work, the authors explored the polar structures in epitaxial BFO superlattices and observed topological objects like bimerons in this system. While similar textures have been observed in other ferroelectric, this is the first time to show such interesting topological objects exist in BFO. I think their result is novel to the field and worthy to be considered in Nature Communications. However, I have the following remarks that need to be adequately addressed before my recommendation of publication.

1. It is not proper to call topological solitons like skyrmions, polar vortex arrays and merons as domains. The polarization field of these topological objects changes continuously in space. Actually, they are more like domain walls or domain defects (see, e.g., a review paper [Rep. Prog. Phys. 80 086501(2017)]). The authors should clarify this.
2. There are confusing statements about the phase of the BFO film. In page 5, it was said that "the low flux at high temperatures can achieve self-regulated growth of tetragonal like (T-like) BFO to thicknesses up to 60 nm with no mixed phase formation". In page 6, it was said that "a peak with narrow horizontal (Q_x) breadth is detected at lower Q_z values, which is indexed as T-like BFO, likely stabilized in the layers closer to the substrate", and that "The measured c/a ratio implies that the BFO is not T-like, but rather moderately strained rhombohedral-like (R-like) due to some degree of strain relaxation." In page 10, it was said that "It is also of note that in the BFO layer, both R-like and T-like regions are identified, in agreement with the experimental observations." So, what is the actual phase of the grown BFO film?
3. Moreover, according to the authors' statement in page 5, the maintaining of a macroscopic strain state and the avoiding of strain relaxation mechanisms is a key to formation of the topological solitons in BFO film. However, in the phase field simulation, strain relaxation with phase separation of T phase and R phase occurs.
4. The c/a value is also not sufficient to judge whether the film is in T phase or R phase.
5. The BFO film thickness is another important factor in the formation of the topological solitons. I notice that in their phase field simulation and effective Hamiltonian simulation, the thickness of the simulated BFO film is at least twice of that in the experiment. The authors should comment about this.

Reviewer #3 (Remarks to the Author):

In recent years, a series of polar topology arrays have been found in ferroelectric PTO/STO superlattices. These polar topologies in PTO may bring many interesting physical properties, such as emergent chiral, local negative permittivity, and conduction properties. It raised numerous attention in physics and materials fields, and some predictable application prospects in microelectronics are expected. It is indeed very exciting to find similar polar topologies in a new system, type-I multiferroics. The related magnetism property will certainly inspire more interesting research in this field. The authors performed solid evidence such as high-quality STEM, EDS, polar map, and PFM images. I suggest the acceptance of this manuscript before solving the following concerns:

1. The author should provide more detailed explanations of why discovering polar solitons in multiferroics is critical. What kind of possible prospects of those polar solitons may differ from the existing ones in the ferroelectric system ?

2. In addition, I still have some concerns. From the STEM images, the polar solitons in BFO are similar to the skyrmions reported in the PTO system. However, as I know, all the skyrmions in PTO should have the same polar structure. Why does BFO system contain such a complex "zoo of ferroelectric solitons"? There must be some fundamental reasons behind this.

3. The polar order also looks like partially exists in the nominally paraelectric STO spacer. This shows that the polar solitons are not completely confined within the BFO layers. What is the stress state in BFO and STO?

4. The polar solitons have been observed in BFO7/STO4 and BFO8/STO4, while the labyrinthine and single domains have been observed in thinner BFO layers. Therefore, could we use phase field simulation or another way to clarify what leads to the stable state of topologies?

This work done by V. Govinden et al. reported the observation of particular ferroelectric topological domains, see ferroelectric solitons, in BFO-STO superlattice sample prepared by PLD. Detailed PFM and TEM results were acquired, and the manuscript was well organized. As the author said, there is an obvious inconsistency between PFM results and TEM results in this paper. Although the author has listed several possibilities, the contradiction between the results cannot be ruled out. The correlation between TEM data and PFM data is worth rethinking, otherwise it is difficult to support the existing conclusions. My main comments are as follows:

1. For this kind of superlattice specimens, plane-view sample was selected to visualize the in-plane polarization distribution. It is well known that TEM samples are usually tens of nanometers thick. Therefore, the electron beam will encounter multiple periodic alternating STO/BFO layers. How to interpret the collected HAADF images is worth discussing. Authors are advised to add STEM simulations to enhance the interpretation of the HAADF images, the existing polarization mapping in Fig.2 is not strong evidence.
2. I can not agree with the authors claimed that the PFM tip enlarge the topological state. The size single topological domain is less than 3nm (line 102), which beyond the detection capability of PFM, there is no so-called "amplification" effect here. At the very least, I don't think it can be used as a reason to explain the gap between PFM and TEM data.

Some suggestions are as follows:

- 1.(line 87) It has been reported that the polar vortex has been observed in BFO multi-iron films, which is similar to the ferroelectric soliton mentioned by the authors. (<https://arxiv.org/pdf/1810.12895v1.pdf>) Can authors relate the topology to the macroscopic physical properties?
- 2.(line 182) In Fig. 1d and e, I suggest author to calibrate the polarization orientation via conventional vector PFM method, and determine the polarization distribution of the skyrmions structure.
3. In line 191, it is no evidence to claim the resolution of PFM cannot resolve domain structures less than 20 nm. The tip radius is less than 10 nm (Rocky Mountain12PT400B), and it is not a problem to resolve domain structures smaller than 20 nm.
- 4.(line 216) The authors claimed that 3D ferroelectric solitons, such as spherical domains, have been observed in BFO/STO superlattice films. Can the 3D structure of ferroelectric solitons be reconstructed according to the in-plane and out-of-plane polarization profiles in Fig.2? And provide more evidences, such as 4D-STEM?
- 5.(line 223) The polar mapping results based on HAADF-STEM show that the size of ferroelectric soliton is only 3 nm, while the size of ferroelectric soliton shown by PFM in FIG. 1d and e is over 20nm. This difference makes me doubt whether PFM and TEM observe the same object. Author are advised to provide HAADF-STEM results, which show multiple soliton configurations or soliton array structure, maybe corresponding 3D-RSM results with apparent periodic structure in plane, similar to Reference 15?
6. In Fig.2, BFO/STO interface is not sharp, including atomic EDS results in Fig. S3. Interface diffusion? or film surface not smooth? From polar mapping, the polarization in STO layer is larger than that in BFO. Can the authors provide more polar mapping results with large scale?
7. In Fig.S2, Can the corresponding topography be provided to exclude the influence of topography on PFM amplitude?

Govinden *et al.* – “Ferroelectric solitons crafted in epitaxial bismuth ferrite superlattices”

Round 1 reports from *Nature Communications*

Reviewer #1 (Remarks to the Author):

This work done by V. Govinden *et al.* reported the observation of particular ferroelectric topological domains, see ferroelectric solitons, in BFO-STO superlattice sample prepared by PLD. Detailed PFM and TEM results were acquired, and the manuscript was well organized. As the author said, there is an obvious inconsistency between PFM results and TEM results in this paper. Although the author has listed several possibilities, the contradiction between the results cannot be ruled out. The correlation between TEM data and PFM data is worth rethinking, otherwise it is difficult to support the existing conclusions. My main comments are as follows:

1. For this kind of superlattice specimens, plane-view sample was selected to visualize the in-plane polarization distribution. It is well known that TEM samples are usually tens of nanometers thick. Therefore, the electron beam will encounter multiple periodic alternating STO/BFO layers. How to interpret the collected HAADF images is worth discussing. Authors are advised to add STEM simulations to enhance the interpretation of the HAADF images, the existing polarization mapping in Fig.2 is not strong evidence.

Author reply:

We agree with the referee that multiple reflections of the electron beam is an important issue. However note that the technique used by us is no different from the STEM imaging techniques used in seminal works by other groups such as Prof. Ramesh and Prof. Muller *et al.*^{1,2}, Prof. Pennycook *et al.*³, Prof. Ma *et al.*⁴ and Prof. Pan *et al.*⁵

In STEM simulations, we first need to construct a supercell model with accurate atomic positions. But it is not possible *a priori* to add topological structural displacements to a standard BFO/STO superlattice model, as we have no pre-confirmed information on which topological structures exist. We cannot determine the exact supercell model for a real superlattice, and typically to capture our “zoo of structures” would require at least thousands of unit cells. Therefore, adding STEM simulations would not shed more light, or help us solve the complicated issue of multiple reflections. This however is a long term and detailed challenge that we intend to pursue and hope to report in the near future.

No changes have been made to our manuscript for this point.

2. I can not agree with the authors claimed that the PFM tip enlarge the topological state. The size single topological domain is less than 3nm (line 102), which beyond the detection capability of PFM, there is no so-called "amplification" effect here. At the very least, I don't think it can be used as a reason to explain the gap between PFM and TEM data.

Author reply:

Thank you for the comment. Topological defects in ferroelectric ultrathin heterostructures are very sensitive to external electrical field. During PFM measurement, an AC bias is necessary for the imaging process, and it also affects the domain size significantly. Our previous experiment shows that even under as low as 150 mV AC bias scan, the domains can be enlarged drastically⁶.

To prove this, we have performed additional phase field simulations. As shown in **Fig. R1**, under an applied electric field of -322 kV/cm, the size of the solitons for $(\text{BFO}_7/\text{STO}_4)_8$ superlattices can be increased from ~ 3 nm to ~ 10 nm for the small solitons, as marked by black arrow. It is also interesting to note that typically smaller solitons are more sensitive to external AC fields.

Fig. R1 | Planar view of the in-plane polarization for $(\text{BFO}_7/\text{STO}_4)_8$ superlattices calculated by phase-field simulations, showing the size-amplification effect under applied field. a) Initial state without external field, b) under an applied vertical electric field of -322 kV/cm. The size of the solitons can be increased with the applied field, for instance in the region denoted by the black arrow.

In addition, we bring to Referee's attention data from our earlier report on the observation of bubbles in PZT films⁶, as shown in **Fig. R2**. One can see that progressive scanning causes tip-induced enlargement of the topological bubbles due to the applied force and AC bias field.

Fig. R2 | Time-resolved domain transition with continuous scanning of PZT_3uc STO films under AC bias of 150 mV, image size: 300 nm × 300 nm (From Ref. ⁶).

Changes made to manuscript:

We have added one sentence on Page 10 of the revised manuscript: “... can in fact enlarge the topological state. Additional phase-field simulations also confirmed this external field effect (See Supplementary Note 8 for phase-field simulation evidence). Third, the removal/thinning of the substrate ...”

Changes made to Supplementary Information:

We have also added Fig. R1 into Supplementary Note 8 (Page 16) (previously Supplementary Note 6), as Fig. S12.

Some suggestions are as follows:

1.(line 87) It has been reported that the polar vortex has been observed in BFO multi-iron films, which is similar to the ferroelectric soliton mentioned by the authors. (<https://arxiv.org/pdf/1810.12895v1.pdf↑#x2191;> Can authors relate the topology to the macroscopic physical properties?

Author reply:

Thank you for this comment. We are indeed aware of this paper reporting polar vortices in BFO/TbScO₃ (BFO/TSO) superlattices, but to the best of our knowledge, we are not aware of its publication. We contacted the senior authors of the arXiv paper, Prof. Schlom and Prof. Ramesh. The reply we received stated that the method used in the arXiv paper was flawed and a much better and more sensitive technique was developed by them. This was published in late 2022 for the same system (BFO/TSO/BFO superlattices) by Carretta *et al.*⁷ In this paper by Carretta *et al.* indeed

multiphase coexistence of polar and antipolar phases was found, and no claim for topological solitons made, but rather, they state: "...shows continuously winding electric dipoles resembling a polarization wave or a series of half-vortices." Note that the macroscopic physical property modifications discussed in Caretta *et al.* is not related to topology, but rather to a phase transformation between polar and non-polar states.

The key difference between the Caretta *et al.* paper and our work is that the former uses DyScO₃ substrates which impose almost zero epitaxial strain on the BFO layer. Our approach of using LAO substrates brings into play the strong effect of epitaxial strain, while the STO layering provides the required electrical boundary conditions for forming topological structures.

Relating the topology to the macroscopic physical properties is indeed an important next step. We are currently in the process of exploring changes such as changes in the local magnetic moment, changes in local conductivity, and also possible changes in the optical response of such soliton samples, the results of which will be reported in a subsequent paper.

No changes have been made to our manuscript for this point.

2.(line 182) In Fig. 1d and e, I suggest author to calibrate the polarization orientation via conventional vector PFM method, and determine the polarization distribution of the skyrmions structure.

Author reply:

Thank you for the suggestion. We have performed such experiments, with the results shown in **Fig. R3**. First, **Figure R3a-e** shows the vertical PFM (V-PFM) and lateral PFM (L-PFM) of a centre divergence soliton structure in the BFO SL. The 3D model (in **Fig. R3a**) depicts the topological defect polarization texture. At the centre and edge of the skyrmion, the dipoles have a strong out of plane component, separated by Néel like walls. This Néel-like domain wall shows a strong in-plane component, pointing outwards from the core. Thus a strong in-plane amplitude is observed at the wall, with opposing phase contrast at the top and bottom hemispheres as shown by their respective arrows. Comparable results were obtained for the centre convergence soliton (**Fig. R3f-j**), with a convergent in-plane component for the Néel wall.

Fig. R3 | Vector PFM showing centre divergence and centre convergence configurations of solitons. (a) Planar TEM and 3-D model of centre divergence domain. Vertical **(b-c)** and lateral **(d-e)** PFM amplitude and phase of a centre divergence domain. **(f)** Planar TEM and 3-D model of centre convergence domain. Vertical **(g-h)** and lateral **(i-j)** PFM amplitude and phase of a centre convergence domain.

Changes made to Supplementary:

We have added **Fig. R3** to the Supplementary Information as a new section **Supplementary Note 5 (Fig. S5)** (on pages 7-8).

Changes made to main manuscript:

We have the following text on **Page 8** of the main manuscript explaining this:

“... either bubble domains⁷ or skyrmions^{8,32}. Examples of vector PFM analysis of single centre-divergent and centre-convergent solitons are presented in Supplementary Note 5. Furthermore ...”

3. In line 191, it is no evidence to claim the resolution of PFM cannot resolve domain structures less than 20 nm. The tip radius is less than 10 nm (Rocky Mountain12PT400B), and it is not a problem to resolve domain structures smaller than 20 nm.

Author reply:

We thank the Referee for this suggestion. We have checked that this tip (Rocky Mountain12PT400B) firstly is only guaranteed with tip radius lower than 20 nm by the manufacturer. More important its stiffness and spring constant indicate it is in the weak indentation regime⁸ and therefore not suitable for contact mode techniques like PFM. Indeed the manufacturer (Bruker) themselves only recommend it for non-contact scanning techniques like EFM and scanning capacitance microscopy.

Second, it is not the tip size alone that matters. In a detailed study, Rodriguez et al.⁹ explained that factors such as contrast transfer function and trailing field effect play a significant influence on the PFM resolution. Indeed, they prove the best obtainable resolution is ~10 nm regime which is what our results show.

Changes made to manuscript:

We have revised the text (**page 8**) to reflect a more accurate value of features we can resolve using PFM, including the reference of Rodriguez et al. as described above:

“The resolution limitations of PFM hinder our ability to image any topological feature smaller than ~10 nm^{7,34}, meaning that this technique alone...”

Where ⁷ = our previous work on bubbles; Zhang *et al.*⁶ and ³⁴ = Rodriguez *et al.*⁹

4.(line 216) The authors claimed that 3D ferroelectric solitons, such as spherical domains, have been observed in BFO/STO superlattice films. Can the 3D structure of ferroelectric solitons be reconstructed according to the in-plane and out-of-plane polarization profiles in Fig.2? And provide more evidences, such as 4D-STEM?

Author reply:

We thank the Referee for this excellent suggestion. We have now reconstructed the 3D structure of the solitons, and a separate figure and discussion have been added (see below). Unfortunately for 4D-STEM, we are sorry it is not within our expertise at the current moment. 4D-STEM data remains under significant controversy in the STEM community, as highlighted by a recent article from Ernst

Ruska Electron Microscopy Centre Forschungszentrum Jülich, one of the world's leading STEM groups¹⁰.

Based on in-plane and out-of-plane polar textures extracted from HAADF-STEM images in Figure 2d-e of the main manuscript, we distinguish three characteristic topological structures, namely skyrmions, bimerons, and merons. The three-dimensional structure models of these solitons are shown in Fig. R4.

Fig. R4 | 3D schematics of topological textures, as deduced from planar and cross-sectional scanning transmission electron microscopy data. (a-c) 3D model of (a) skyrmion, (b) bimeron and (c) meron based on in-plane and out-of-plane polar maps in Fig. 2 of the main manuscript.

The in-plane HAADF-STEM images of various types of solitons are given in Fig. R5d. The in-plane views (I , centre-convergent polar; II , centre-divergent polar), the cross-sectional view (III, anti-parallel (up vs down) polar) are given in Fig. R5a. The in-plane view of a meron and cross-sectional view of a bimeron are given in Fig. R5b and Fig. R5c, respectively. Note that in Fig. R5d, the in-plane polar map reveals the coexistence of multiple soliton configurations, containing centre-convergent polar, anti-vortex, centre-divergent polar and bimeron. Such an observation is consistent with the PFM images shown in the main manuscript, where larger soliton-like structures are identified (also see our response to Point 5, next).

Changes made to Supplementary Information:

We have combined Fig. R4 and evidence for coexistence of multiple soliton configurations (given below in Fig. R5d) as a new supplementary figure (Fig. S7) in Supplementary Note 6 (page 10).

Changes made to manuscript:

We have added the following text (**Page 9**) in the main manuscript:

“... reveal various topological states, as further detailed in Supplementary Note 6. In Figs. 2d ...”

Fig. R5 | Coexistence of multiple soliton configurations. (a-c) 3D polar models of (a) skyrmion, (b) meron, and (c) bimeron. (a) shows the top plane (centre-convergent polar), bottom plane (centre-divergent polar) and cross-sectional view of the skyrmion. (d) In-plane polar map containing centre-convergent polar (I), anti-vortex (II), centre-divergent polar (III) and bimeron (IV) textures. Scale bar, 1nm.

5.(line 223) The polar mapping results based on HAADF-STEM show that the size of ferroelectric soliton is only 3 nm, while the size of ferroelectric soliton shown by PFM in FIG. 1d and e is over 20nm. This difference makes me doubt whether PFM and TEM observe the same object. Author are advised to provide HAADF-STEM results, which show multiple soliton configurations or soliton array structure, maybe corresponding 3D-RSM results with apparent periodic structure in plane, similar to Reference 15?

Author reply:

We thank the Referee for this insightful point. The discrepancy between the size of topological features observed by PFM and STEM has also been discussed in the well-cited and highly regarded work of Yadav et al.¹¹ (Ref. 16 in our revised main manuscript). We highlight that in PTO/STO, the researchers observed regular polar vortex (Ref. 16) and skyrmion (Ref. 8) arrays, as well as corresponding in-plane periodicity. This allowed them to detect satellite peaks in the x-ray diffraction RSMs.

However, in our BFO/STO system, due to the co-existence of multiple solitons (see **Fig. R5d**) with no regular periodic array, the reciprocal space will not show any satellite peaks. To prove this, we have performed RSMs with various azimuthal (ϕ) angle for the BFO/STO superlattice sample (**Fig. R6**). Regarding a 3D-RSM to search for periodic in-plane domain structures, we have made significant attempts to obtain such data, with no evidence of periodic structures or superlattice peaks. We believe the reason for this is the lack of periodic structures – in other words, although there are soliton-like textures observed in the samples, they are not of the strongly periodic nature shown in Reference 16 where they are perfectly aligned in crystallographic orientations.

We attribute this phenomenon to the wider range of degrees of freedom in the BFO structure – namely that there can be various strain states (viz., T-like and R-like BFO) as well as a high level of disorder, as evidenced by the very wide XRD peak in Fig. 1b.

Fig. R6 | Symmetrical RSMs around the 002 reflection for different azimuthal angles. (a) A 3D representation of the four RSMs taken at different angles. (b-e) RSMs around the 002 reflection for a $(\text{BFO}_7\text{-STO}_4)_{10}$ superlattice at different azimuthal (ϕ) angles, showing no evidence of superstructure peaks in the horizontal (in plane) directions.

We absolutely agree that the PFM is not imaging individual solitons. Rather we believe the density of solitons is such that the PFM is rendering an image of a cluster. To prove this, we have shown STEM images showing multiple soliton configurations as requested by the referee, please see **Fig. R5d**, and the respective changes made to the manuscript.

Changes made to Supplementary Information:

We have added evidence that shows the solitons exist in multiple configurations into **Supplementary Note 6** (page 10).

6. In Fig.2, BFO/STO interface is not sharp, including atomic EDS results in Fig. S3. Interface diffusion? or film surface not smooth?

Author reply:

We thank the Referee for this question, which raises a crucial point. The interface roughness is a reflection of the strong polarization strain coupling – i.e., when the polarization curls, we expect this to be accompanied by a distortion of the unit cell. It is definitely not interface diffusion- using the EDS mapping results, we can clearly read 7 layers of Fe atoms and 4 layers of Ti atoms atomic level sharpness (see **Fig. S3** in the Supplementary Information).

We highlight the same issue is seen by the breakthrough works of Tang *et al.*¹² and Yadav *et al.*¹¹. We have reproduced the atomic resolution map at the interface of PTO and STO from the former work (**Fig. R7**), which shows a very strong undulation at the interface as a result of the polarization strain coupling.

Fig. R7 | Published STEM data on PTO/STO superlattices¹². (top - D) Low-magnification high resolution HAADF-STEM image and (bottom - E) GPA analysis of the STEM data reveals out-of-plane strain.

From polar mapping, the polarization in STO layer is larger than that in BFO. Can the authors provide more polar mapping results with large scale?

To address this question, we performed the polarization mapping at larger scale. We find that although polarization also partially exists in the STO layer, the polarization in the BFO is still significantly greater than in the STO (**Fig. R8c,f**). Polarization maps of Figure 2e (with a larger scale) are shown in **Fig. R8b,e**.

Fig. R8 | Polarization in BFO and STO layers. (a,d) HAADF-STEM images of BFO/STO layers. (b,e) Polar map of HAADF-STEM images. (c,f) Polarization of BFO and STO layers. The colour maps represent the magnitude of polarization. Scale bar, 2 nm.

Changes made to Supplementary Information:

We have now included an additional section in the Supplementary Information (**Supplementary Note 6**) (page 11), explaining the results of **Fig. R8**.

7. In Fig.S2, Can the corresponding topography be provided to exclude the influence of topography on PFM amplitude?

Author reply:

Thank you for this comment. We have acquired topography for all our PFM images, and we have reproduced some examples in **Fig. R9**. For these images, we can see that there is no clear correlation between the topography and the PFM amplitude.

Fig. R9 | Piezoresponse force microscopy (PFM) scans for three different $[7/4]_{10}$ BFO/STO superlattices. Topography (left column), PFM amplitude (middle column), and PFM phase (right column). Notably, all the samples show small circular like domains, and there is no discernible correlation between the PFM amplitude and the topography.

Changes made to manuscript:

No changes. If the Referee requests, we can include **Fig. R9** in the Supplementary Information.

Reviewer #2 (Remarks to the Author):

Over the past two decades, polar topological structures in ferroelectrics have attracted intensive attention for their potential as the building blocks in developing nanoelectronics. In this work, the authors explored the polar structures in epitaxial BFO superlattices and observed topological objects like bimerons in this system. While similar textures have been observed in other ferroelectric, this is the first time to show such interesting topological objects exist in BFO. I think their result is novel to the field and worthy to be considered in Nature Communications. However, I have the following remarks that need to be adequately addressed before my recommendation of publication.

1. It is not proper to call topological solitons like skyrmions, polar vortex arrays and merons as domains. The polarization field of these topological objects changes continuously in space. Actually, they are more like domain walls or domain defects (see, e.g., a review paper [Rep. Prog. Phys. 80 086501(2017)]). The authors should clarify this.

Author reply and changes to the manuscript:

Thank you for the comment, and we apologize for this oversight. Indeed, we agree that these structures should not be called domains. That said, we continue to use the accepted nomenclature in the literature, for instance regarding “bubble domains” and “spherical domains”. We have however removed all reference to “domains” when referring to our soliton polarization textures.

Changes made to manuscript:

We have removed all mention of domains, replacing it with “polarization textures”. These changes have been highlighted in the revised (marked up) manuscript.

2. There are confusing statements about the phase of the BFO film. In page 5, it was said that "the low flux at high temperatures can achieve self-regulated growth of tetragonal like (T-like) BFO to thicknesses up to 60 nm with no mixed phase formation". In page 6, it was said that "a peak with narrow horizontal (Q_x) breadth is detected at lower Q_z values, which is indexed as T-like BFO, likely stabilized in the layers closer to the substrate", and that "The measured c/a ratio implies that the BFO is not T-like, but rather moderately strained rhombohedral-like (R-like) due to some degree of strain relaxation." In page 10, it was said that "It is also of note that in the BFO layer, both R-like and T-like regions are identified, in agreement with the experimental observations." So, what is the actual phase of the grown BFO film?

Author reply:

Thank you for the important comment, and we apologize for the confusion. The complication here is that the BFO film is in a multiphase state. Indeed, when the system shows a soliton state, with significant polarization curling, one could argue that many structural phases (tetragonal, monoclinic,

rhombohedral, orthorhombic) could be present. This is clearly shown in the multiple polarization states observed in **Fig. R5d**.

To gain further insight into these possible structural phases, we have performed a more detailed GPA analysis, as explained below.

Throughout the film, as a general rule, the BFO layers are under out-of-plane tensile strain, consistent with the in-plane compressive stress imposed by the LAO substrate. As shown in HAADF-STEM image and GPA analysis in **Fig. R10**, although the atomic interface between BFO and STO is clean (**Fig. R10a**), the tensile strain also partially exists in the STO layer, leading to induced polar order in the STO. The in-plane strain appears to be uniform in BFO and STO. The value depends on the lattice mismatch between the superlattice and the substrate.

Fig. R10 | GPA analysis of BFO/STO superlattice near the substrate-film interface. (a) HAADF-STEM image of BFO/STO film. **(b-d)** In-plane strain (ϵ_{xx}), out-of-plane strain (ϵ_{yy}) and shear strain (ϵ_{xy}) distribution of the film. **(e)** Average in-plane, out-of-plane, shear strain as a function of distance from the substrate.

Moreover, we observe periodic dislocations at the interface between the bottom BFO and LSMO, where the positions can be identified through the large in-plane strain variations (see **Fig. R10b**). **Figure R10b-d** shows the in-plane strain (ϵ_{xx}), out-of-plane strain (ϵ_{yy}) and shear strain (ϵ_{xy}) distribution of the film. However note that no dislocations are evident in the BFO/STO superlattice layers. The in-plane strain in the middle BFO/STO area increases monotonically with distance from the substrate. At the same time, the out-of-plane strain and shear strain do not show a correlation with this change. **Figure R10e** shows the relationship between average in-plane, out-of-plane, and

shear strain with the distance from the substrate. We can see the BFO/STO layers in a state of “strain glass” where there is significant local strain variations on the nanometre scale but this does not travel beyond a few nm. In other words, the BFO/STO layers host multiple local strain states which translates into a multiphase co-existence.

To make our argument further clear we also show the strain and polarization map from a region taken in the middle of the BFO/STO superlattice (**Fig. R11**). There is a large range of strain variation, typically for such values we would expect the origins to be either dislocation formation or chemical disorder. We have already shown that there are no dislocations and the interfaces for the BFO/STO layers are sufficiently sharp to not allow such variations occurring over several nanometres. The only other possibility is that the local polarization pattern is changing dramatically, i.e. it is no longer constrained to the (001) (for T-like) or (111) (for R-like) BFO. It is assuming rotations that are not constricted to well-known crystallographic states (or strained phases) in order to form the solitons. As a result, we cannot use the traditional understanding of thin films applied to BFO that the film is in a *single* T or R phase. It houses a plethora of strain states due to the zoo of solitons formed.

Fig. R11 | Strain and polarization analysis of BFO/STO film in the centre of the superlattice. (a) HAADF-STEM image in the middle layers of a $(7/4)_{10}$ BFO/STO superlattice. **(b)** Polar map of (a). **(c-e)** In-plane strain (ϵ_{xx}), out-of-plane strain (ϵ_{yy}) and shear strain (ϵ_{xy}) distributions of the superlattice. Scale bar, 2nm.

Our comment on page 5 “the low flux at high temperatures can achieve self-regulated growth of tetragonal like (T-like) BFO to thicknesses up to 60 nm with no mixed phase formation” was included simply to explain to the reader that our unique PLD chamber provides the perfect conditions to create thick T-like BFO that is not possible in other labs.

Changes made to Supplementary Information:

We have added **Fig. R10** and **Fig. R11** to the Supplementary Information, as **Supplementary Note 7** (pages 12-14).

Changes made to manuscript:

(Page 7) We have changed to read:

“Later we will compare this value with the results from local measurements of the lattice parameters using STEM. ~~The measured c/a ratio implies that the BFO is not T-like, but rather moderately strained rhombohedral-like (R-like) due to some degree of strain relaxation.~~ We will show that the local strain mapping reveals that the BFO layers are neither pure T nor pure R, but rather a mixture of strain and polar states. This multi-strain state is key to relax the excess depolarization and elastic energies in this superlattice system. ~~This specific strain state is key, as we will see later when we discuss first-principles-based computations.~~”

We have also added a short description of Supplementary Note 7 on page 9, as follows:

“...smaller characteristic size (~ 3.5 nm) (Fig. 2b) as compared to those in the PTO/STO system (8 nm)⁸. Moreover, note that Fig. 2c reveals a range of c/a ratios with an average of 1.054, which implies implying that the BFO layers ~~is~~ are neither pure T-like, nor pure R-like³⁶, albeit with a slightly larger c/a than previously reported for typical R-like BFO³⁶. To understand the origins of such values, we carried out detailed local strain geometric phase analysis (GPA) mapping results (full details in Supplementary Note 7). First, the GPA confirms the lack of any dislocation cores across the entire BFO/STO superlattice layers, and we do not see any obvious signs of chemical disorder. The observed wide range of c/a thus should stem from an alternate relaxation mechanism. Noting that the strain mapping shows the superlattices to be in a state of “strain glass,” i.e., no long range strain order but consistent with the size scales of the multiple topological solitons (as seen in Fig. 2d and S6) we propose that the superlattices host a plethora of strain states (and hence ferroelectric phases) due to the coexistence of a multiple ferroelectric polarization patterns. ~~This is once again consistent with the moderately high level of in-plane residual strain imposed by the substrate.~~”

So far, we have shown through PFM and STEM imaging...”

3. Moreover, according to the authors' statement in page 5, the maintaining of a macroscopic strain state and the avoiding of strain relaxation mechanisms is a key to formation of the topological solitons in BFO film. However, in the phase field simulation, strain relaxation with phase separation of T phase and R phase occurs.

Author reply:

We appreciate this valuable comment. In fact, the misfit strain between LAO and BFO bulk is quite large (~4.4 %), so strain relaxation is unavoidable. The key for the formation of the topological solitons in BFO film is not to completely avoid strain relaxation, but to control the degree of strain relaxation through control of growth conditions.

In our case, strain relaxation does not occur through misfit dislocation formation, but rather through the formation of solitons. The GPA analysis (see Point 2 of Referee 2 above; **Fig. R10**) shows that misfit dislocations do not occur in the BFO-STO layers, however there are several dislocations at the LAO-LSMO interface [**Fig. R10(b)**]. This clearly demonstrates that the elastic energy is relaxed through the formation of topological solitons in the BFO-STO layers. The phase-field simulations indicate that topological solitons can be stabilized only when the substrate-imposed strain is below about -1.7%, consistent with the experimental observations. Moreover, from phase-field simulations, we have discovered that the majority of the BFO layer is still R phase, while some T-like phase (more like monoclinic phase) can also be seen near the soliton walls, as shown in Fig. 3f of the manuscript.

Changes made to manuscript:

Please refer to our detailed discussion on the strain state of our superlattices, i.e., our **reply to point 2 of Referee 2** above.

4. The c/a value is also not sufficient to judge whether the film is in T phase or R phase.

Author reply:

We agree with the Referee, but we point out that the film is not a single R-like or T-like phase, but a range of different phases due to the presence of these soliton structures, as discussed in our **reply to point 2 of Referee 2** above.

5. The BFO film thickness is another important factor in the formation of the topological solitons. I notice that in their phase field simulation and effective Hamiltonian simulation, the thickness of the simulated BFO film is at least twice of that in the experiment. The authors should comment about this.

Author reply:

We thank the reviewer for the valuable comment. In our phase-field simulation, the thickness of the simulated BFO and STO layers is the same as that of the experiment, both of which are BFO₇/STO₄.

Regarding the effective Hamiltonian calculations, we have also carried out simulations with BFO film thickness similar to the experiment. The results (**Fig. R12**) show similar vortex features at the same location as for higher thickness studied in the paper, with similar Pontryagin's charge density. We did not show the results for this thickness in the paper as the features of the vortex were not easily visible to the naked eye. To clarify further, we have added text in the section on effective Hamiltonian calculations.

Changes made to manuscript:

We added one sentence on **Page 19** (in the Methods for Phase Field Simulations) of the revised manuscript to emphasize this point: "... on an LaAlO₃ substrate. In the film layer, 7 unit cells of BFO and 4 unit cells of STO are deposited periodically, which is consistent with the experiment. Two sets of order parameters ..."

We have also added the following text (page 12) in the main manuscript:

"To further strengthen our predictions of the formation of topological objects under compressive strain, we also used a first-principles-based effective Hamiltonian¹³ to study thin film BFO. A 16 u.c. thick (001) oriented BFO film with initial 109° domain structure was placed under mechanical boundary conditions of epitaxial compressive strain varying up to -5%, with open-circuit-like electrical boundary conditions. A thickness of 16 unit cells was chosen to better show the vortex features. However, thickness equal to the experiments also showed similar polar mode features (results not shown here). Multidomain structures are known to be preferred over monodomains under open circuit like electrical boundary conditions¹²."

Fig. R12 | Polar mode in the x-z plane for BFO films with thickness = 6 unit cells a) and 8 unit cells b). Vortex-like features are visible in both cases.

Reviewer #3 (Remarks to the Author):

In recent years, a series of polar topology arrays have been found in ferroelectric PTO/STO superlattices. These polar topologies in PTO may bring many interesting physical properties, such as emergent chiral, local negative permittivity, and conduction properties. It raised numerous attention in physics and materials fields, and some predictable application prospects in microelectronics are expected. It is indeed very exciting to find similar polar topologies in a new system, type-I multiferroics. The related magnetism property will certainly inspire more interesting research in this field. The authors performed solid evidence such as high-quality STEM, EDS, polar map, and PFM images. I suggest the acceptance of this manuscript before solving the following concerns:

1. The author should provide more detailed explanations of why discovering polar solitons in multiferroics is critical. What kind of possible prospects of those polar solitons may differ from the existing ones in the ferroelectric system?

Author reply:

Thank you for the comment. We believe that finding solitons and other topological structures in a type-1 multiferroic is an important discovery for the following reasons:

- First, from a fundamental point of view, the observation of exotic order-parameter topologies represents an exciting horizon of modern condensed matter physics. The complex evolution in controlled phase space is only just being explored. This work could chart the course to the creation of engineered multiferroic topologies which show the coexistence of polar and spin topologies in a single material, enabling functionalities such as ultra-fast electric-field control of magnetism, light, etc.
- Solitons in purely ferroelectric systems can only respond to electrical and/or optical stimulus, whereas multiferroic solitons can interact with electrical, magnetic, and optical fields;
- The magnetoelectric coupling that exists in BFO can open up paths towards a topological memory element that can be tuned by electric field, but then store the memory state in the magnetic/electric order parameter;
- All these functionalities are possible at room temperature, and does not rely on finding coupling at cryogenic temperatures;
- The richness of physics in this material means that other discoveries may await – for example, what happens to the complex electronic structure within these solitons? Does the optical band gap and the local conductivity change? Could we engineer a high-density array of solitons where the local properties of the solitons (modified transition temperatures, different structural) dictates the bulk response of the system?

These points were alluded to in the Outlook section of our manuscript; however, we have now added a summary of the above in the introduction, as per the Referee's comment.

Changes made to manuscript:

We have made the following changes in the Introduction of the manuscript to highlight these virtues of multiferroic solitons:

Introduction (Page 4):

"... both fundamentally and practically. Whilst engineered epitaxial BFO ..." "... is still to be achieved. Demonstration of solitons in multiferroic BFO offer exciting prospects over and above their purely ferroelectric counterparts, including new local spin-related physics, and novel ways to engineer spin-lattice coupling. The coexistence of both polar and spin topologies in a single material would enable functionalities such as ultra-fast electric-field control of magnetism, strain, magnetostriction, etc. Moreover, the room-temperature coupling inherent to these systems make them promising candidates for new spintronic devices. A natural question thus arises ..."

2. In addition, I still have some concerns. From the STEM images, the polar solitons in BFO are similar to the skyrmions reported in the PTO system. However, as I know, all the skyrmions in PTO should have the same polar structure. Why does BFO system contain such a complex "zoo of ferroelectric solitons"? There must be some fundamental reasons behind this.

Author reply:

Thank you for making this important point. Indeed, it is critical to understand why our system does display such a wide array of different topological structures.

We believe the explanation lies in the wide array of strain and crystallographic structures that are possible in BFO. Depending on the imposed epitaxial strain, BFO can form in either R-like or T-like phases, and we observe evidence of both phases (albeit with a very tiny phase fraction of the T-like phase) from our XRD measurements (Fig. 1b). Recently we showed that through anisotropic strain, one can stabilize a new triclinic phase of BFO. This proves that the material is rather compliant and "malleable" for strain and crystallographic structures.

A further important distinction between the BFO and PTO systems is the fact that BFO is rhombohedral in bulk (i.e., with 3 cartesian components of the polarization) while PZT with large Ti composition or PTO is tetragonal in bulk (i.e., with only one cartesian component of the polarization). It is quite likely that this fact makes BFO richer for ferroelectric solitons simply because there are more possible orientations of the polarization vector.

Finally, the electric dipoles of BFO also compete or interact with oxygen octahedral tilting and magnetism¹⁴, whereas this does not occur in PTO. We believe that it is possible that this makes BFO richer for topological defects.

Changes made to manuscript:

We have included an additional discussion at **Page 10 (Page 11 in Highlighted version)** section to this effect:

“... scanning field⁷. The fact that we observe such a wide array of topological structures in the BFO/STO superlattices, whereas in the PTO-STO system only a single type of polar skyrmion is observed¹⁶, is likely related to the increased degrees of freedom in the BFO system (*cf.* the multitude of structural phases evidenced by GPA analysis in Supplementary Note 7). Since BFO is rhombohedral in bulk (i.e., with three cartesian components of the polarization) while PTO is tetragonal in bulk (i.e., with only one cartesian component of the polarization), the polarization vector in the former case has more freedom to form various types of polar arrangements. Moreover, the fact that the electric dipoles in BFO compete/interact with the oxygen octahedral rotations and the magnetic order parameter (while this does not occur in PTO) likely makes the BFO system richer for topological defects.”

3. The polar order also looks like partially exists in the nominally paraelectric STO spacer. This shows that the polar solitons are not completely confined within the BFO layers. What is the stress state in BFO and STO?

Author reply:

Thank you for this important comment. We have indeed noticed that the polar order extends beyond the BFO layers and into the STO. We discuss this on page 8, where we attribute the effect to the strong electrostatic coupling which causes the STO to become polar. This is further evidenced in the extracted *c/a* ratios for the BFO and STO layers (displayed in Fig. 2c and reproduced in Fig. **R13**), as well as the more detailed GPA analysis of the strain state in the various layers (see response to point 2 of Referee 2 and Fig. **R10** and Fig. **R11**).

From our XRD data, particularly the RSMs given in Fig. S4, we obtain the following ranges of lattice parameters for the “averaged” BFO-STO film (note that the superlattice shows a broad peak which is the combined diffraction from the BFO and STO layers):

$$c_{\text{BFO-STO}} = 4.007 \pm 0.018 \text{ \AA}$$

$$a_{\text{BFO-STO}} = 3.91 \pm 0.08 \text{ \AA}$$

Also note that the LSMO is partially relaxed, as observed in our previous studies¹⁵.

Fig. R13 | (Fig. 2c in the main manuscript) – measured c/a ratios for the BFO and STO layers, extracted from cross sectional STEM images. Note that the nominally cubic STO becomes tetragonal from the imposed strain, with an average c/a ratio of about 1.035.

Please also refer to the response to points 2 and 3 of Referee 2, where we discuss GPA results.

Changes made to manuscript:

The GPA analysis has been added to the Supplementary Information as **Supplementary Note 7**. We also agree that lines 168-172 (in the original submitted version) were confusing and so they have been replaced with text explaining the multitude of strain states – please see the new text in our **reply to point 2 of Referee 2** above.

4. The polar solitons have been observed in BFO7/STO4 and BFO8/STO4, while the labyrinthine and single domains have been observed in thinner BFO layers. Therefore, could we use phase field simulation or another way to clarify what leads to the stable state of topologies?

Author reply:

We appreciate this valuable comment on the mechanisms for the differences of topological domain evolution in BFO/STO superlattice heterostructures with varying layer thicknesses. Following this suggestion, we have employed the phase-field simulations to understand the thermodynamics of the phase stability in BFO/STO superlattices.

The film thickness phase diagram of the BFO_n/STO₄ superlattices is first established (**Fig. R14**). When the BFO layer thickness is large ($n \geq 12$), 71° twin domains are formed, similar to the domain structure for BFO thin films on a STO substrate¹⁶. Upon decreasing the BFO layer thickness ($5 \leq n < 11$), the polar soliton state emerges, which is consistent with the experimental observations. When the BFO layer thickness is further reduced ($n \leq 4$), a monodomain state is the ground state, as has been shown in Figure S3. To understand the energetics for the phase transitions, the individual energy densities vs. film thickness n was also plotted. The bulk energy density of the BFO/STO superlattices increases with the decrease of film thickness, due to the large reduction of the spontaneous polarization with

thinner film. The elastic energy density, however, reduces dramatically which could compensate for the increase of the bulk energy density. The gradient and electric energy densities only change slightly with varying thickness. It can be concluded that the main driving force of the topological phase transitions is the competition between bulk and elastic energy densities.

Fig. R14 | Film thickness phase diagram for $\text{BFO}_n/\text{STO}_4$ superlattices from phase-field simulations. The change of the individual energy densities with film thickness n (with respect to energy densities of $\text{BFO}_7/\text{STO}_4$ superlattice) is also shown.

Summary of changes made to manuscript:

1. We have removed all reference to “domains” when referring to our soliton polarization textures in the revised manuscript:

“This phenomenon enables the formation of exotic ~~domain structures~~ **polarization textures** which...” (Page 3)

“In this context, the observation of spherical and transitional topologies such as skyrmions, polar vortex arrays, merons and ~~bubble domains~~ **electrical bubbles** in BFO...” (Page 4)

“Next, we discuss the **non-trivial** ferroelectric topologies ~~domain structure~~, imaged using...” (Page 8)

“These ~~nanodomains~~ **polarization textures** show blurry amplitude contrast and a faint ~~upward domain phase reversal at~~ of the domain wall.” (Page 8)

“In Figs. 2d, I and II, we observe both in-plane centre-divergent and centre-convergent polar textures, which take the **circular** form ~~of circular domains in...~~” (Page 9)

“A “bimeron,” comprising both centre-divergent and anti-vortex polar structures, is also identified (Fig. 2d, III). This structure appears as the fusion of two **bubbles** ~~domains~~ in the STEM image of Fig. 2a.” (Page 9)

“...which had (i) a vortex in the x-z plane, and (ii) convergent/divergent ~~domain walls~~ **polar textures** in the x-y plane located at the circumference of the vortices.” (Page 12-clean version, and Page 13-highlighted version)

“PFM amplitude and phase images, revealing **complex non-trivial topologies** ~~a topological-like domain texture.~~” (Page 24)

“Enlarged cross-sectional STEM-HAADF images and corresponding polar vectors, showing (I) **an** ~~domain with~~ anti-parallel (up-down) polarization, and (II) a trapezoidal ~~domain~~ **shape** with convergent polar configuration.” (Page 25)

2. We have added three sentences on Page 4 of the revised manuscript:

“Whilst engineered epitaxial BFO heterostructures have shown writable vortex cores²⁶, centre convergent and quad-domain structures²⁷ or self-assembled flux closure arrays⁵, the observation of topological solitons is still to be achieved. **Demonstration of solitons in multiferroic BFO offer exciting prospects over and above their purely ferroelectric counterparts, including new local spin-related physics, and novel ways to engineer spin-lattice coupling. The coexistence of both polar and spin topologies in a single material would enable functionalities such as ultra-fast electric-field control of magnetism, strain, magnetostriction, etc. Moreover, the room-temperature coupling inherent to these systems make them promising candidates for new spintronic devices.**”

3. We have changed one sentence on Page 6 of the revised manuscript:

“The thickness of the BFO layer is pivotal: each layer must be thin enough to maintain the imposed “macroscopic” strain and dipolar coupling at the interface, **without misfit dislocation formation**. At the same time, it must also allow ~~while also~~ coupling across the STO spacers, ~~but not to be locally~~ **without being** influenced by intrinsic size effects (Fig. 1a).”

4. We have deleted two sentences and added two sentences on **Page 7** of the revised manuscript:

~~“The measured c/a ratio implies that the BFO is not T-like, but rather moderately strained rhombohedral like (R-like) due to some degree of strain relaxation. We will show that the local strain mapping reveals that the BFO layers are neither pure T nor pure R, but rather a mixture of strain and polar states. This multi-strain state is key to relax the excess depolarization and elastic energies in this superlattice system. This specific strain state is key, as we will see later when we discuss first principles based computations.”~~

5. We have added one sentence on **Page 8** of the revised manuscript:

“Previously, such features were ascribed to either bubble domains⁷ or skyrmions^{8,32}. Examples of vector PFM analysis of single centre-divergent and centre-convergent solitons are presented in Supplementary Note 5.”

6. We have changed one sentence on **Page 8** of the revised manuscript:

~~“The resolution limitations of PFM hinder our ability to image any topological feature smaller than ~ 10 nm^{7,34}, meaning that...”~~

7. We have added one sentence on **Page 9** of the revised manuscript:

“The planar view HAADF-STEM image and corresponding vector displacement mapping reveal various topological states, as further detailed in Supplementary Note 6.”

8. We have added four sentences on **Page 9-10 (Page 10 in highlighted version)** of the revised manuscript:

~~“Moreover, note that Fig. 2c reveals an average a c/a ratios of with an average of 1.0540, implies implying that the BFO layers are neither pure T-like, nor pure R-like albeit with a slightly larger c/a than previously reported for typical R-like BFO³⁶. To understand the origin of such values, we carried out detailed local strain geometric phase analysis (GPA) mapping results (full details in Supplementary Note 7). First, the GPA confirms the lack of any dislocation cores across the entire BFO/STO superlattice layers, and we do not observe any obvious signs of chemical disorder. The observed wide range of c/a thus should stem from an alternate relaxation mechanism. Noting that the strain mapping shows the superlattices to be in a state of “strain glass,” i.e., no long range strain order but consistent with the size scales of the multiple topological solitons (as seen in Fig. 2d and S6) we propose that the superlattices host a plethora of strain states (and hence ferroelectric phases) due to the coexistence of a multiple~~

ferroelectric polarization patterns. This is once again consistent with the moderately high level of in-plane residual strain imposed by the substrate.”

9. We have added one sentence on **Page 10** of the revised manuscript:

“Second, we point out that during PFM imaging, the trailing field effect of the tip¹⁴, through the applied slight pressure and electric field, can in fact enlarge the topological state. Additional phase-field simulations also confirmed this external field effect (See Supplementary Note 8 for phase-field simulation evidence).”

10. We have added three sentences on **Page 10-11 (Page 11 in highlighted version)** of the revised manuscript:

“We have, conversely, shown that bubbles are extremely sensitive to applied scanning probe microscopy pressure and scanning field⁷. The fact that we observe such a wide array of topological structures in the BFO/STO superlattices, whereas in the PTO-STO system only a single type of polar skyrmion is observed¹⁶, is likely related to the increased degrees of freedom in the BFO system (*cf.* the multitude of structural phases evidenced by GPA analysis in Supplementary Note 7). Since BFO is rhombohedral in bulk (i.e., with three cartesian components of the polarization) while PTO is tetragonal in bulk (i.e., with only one cartesian component of the polarization), the polarization vector in the former case has more freedom to form various types of polar arrangements. Moreover, the fact that the electric dipoles in BFO compete/interact with the oxygen octahedral rotations and the magnetic order parameter (while this does not occur in PTO) likely makes the BFO system richer for topological defects.”

11. We have added three sentences on **Page 12** of the revised manuscript:

“...both R-like and T-like regions are identified, in agreement with the experimental observations. In addition, the equilibrium structure in BFO_n/STO₄ superlattice heterostructures with varying BFO layer thicknesses was studied by phase-field simulations. It was discovered that when the BFO layer thickness decreases from 28 unit cells, the BFO layer undergoes a topological phase transition from twin domains (for $n > 11$) to solitons (n between 5 to 10) to monodomain ($n < 5$). The main driving force of this topological phase transitions is the competition between bulk and elastic energy densities, as shown in Fig. S13 (Supplementary Note 8).”

12. We have added one sentence on **Page 12** of the revised manuscript:

“...with open-circuit-like electrical boundary conditions. A thickness of 16 unit cells was chosen to better show the vortex features. However, thickness equal to the experiments also showed similar polar mode features (results not shown here).”

13. We have added one sentence on **Page 19** of the revised manuscript:

“...grown on an LaAlO_3 substrate. In the film layer, 7 unit cells of BFO and 4 unit cells of STO are deposited periodically, which is consistent with the experiment.”

Summary of changes made to Supplementary Information:

1. A new **Supplementary Note 5**.
We have added **Fig. R3** to **Supplementary Note 5**.
2. We have added **Fig. R5** and **Fig. R8** to **Supplementary Note 6**.
3. A new **Supplementary Note 7**.
We have added **Fig. R10** and **Fig. R11** to **Supplementary Note 7**.
4. We have added **Fig. R1** and **Fig. R14** to **Supplementary Note 8**.

Reply to Referees References

1. Das, S. *et al.* Observation of room-temperature polar skyrmions. *Nature* **568**, 368–372 (2019).
2. Das, S. *et al.* Local negative permittivity and topological phase transition in polar skyrmions. *Nat Mater* **20**, 194–201 (2021).
3. Yin, J. *et al.* Nanoscale bubble domains with polar topologies in bulk ferroelectrics. *Nat Commun* **12**, 3632 (2021).
4. Wang, Y. J. *et al.* Polar meron lattice in strained oxide ferroelectrics. *Nat Mater* **19**, 881–886 (2020).
5. Han, L. *et al.* High-density switchable skyrmion-like polar nanodomains integrated on silicon. *Nature* **603**, 63–67 (2022).
6. Zhang, Q. *et al.* Nanoscale Bubble Domains and Topological Transitions in Ultrathin Ferroelectric Films. *Advanced Materials* **29**, 1702375 (2017).
7. Caretta, L. *et al.* Non-volatile electric-field control of inversion symmetry. *Nat Mater* **22**, 207–215 (2022).
8. Kalinin, S. V & Bonnell, D. A. Imaging mechanism of piezoresponse force microscopy of ferroelectric surfaces. *Physical Review B* **65**, 125408 (2002).
9. Kalinin, S. V. *et al.* Spatial resolution, information limit, and contrast transfer in piezoresponse force microscopy. *Nanotechnology* **17**, 3400 (2006).
10. Strauch, A. *et al.* Systematic Errors of Electric Field Measurements in Ferroelectrics by Unit Cell Averaged Momentum Transfers in STEM. *Microscopy and Microanalysis* (2023) doi:10.1093/micmic/ozad016/7055859.
11. Yadav, A. K. *et al.* Observation of polar vortices in oxide superlattices. *Nature* **530**, 198–201 (2016).
12. Tang, Y. L. *et al.* Observation of a periodic array of flux-closure quadrants in strained ferroelectric PbTiO₃ films. *Science* **348**, 547–551 (2015).
13. Prosandeev, S., Wang, D., Ren, W., Íñiguez, J. & Bellaiche, L. Novel Nanoscale Twinned Phases in Perovskite Oxides. *Adv Funct Mater* **23**, 234–240 (2013).
14. Diéguez, O., González-Vázquez, O. E., Wojdeł, J. C. & Íñiguez, J. First-principles predictions of low-energy phases of multiferroic BiFeO₃. *Physical Review B* **83**, 94105 (2011).
15. Arredondo, M. *et al.* Chemistry of Ruddlesden-Popper planar faults at a ferroelectric-ferromagnet perovskite interface. *Journal of Applied Physics* **109**, 84101 (2011).
16. Nakashima, S. *et al.* Bulk photovoltaic effect in a BiFeO₃ thin film on a SrTiO₃ substrate. *Japanese Journal of Applied Physics* **53**, 09PA16 (2014).

REVIEWER COMMENTS

Reviewer #1 (Remarks to the Author):

As far as I know, there has been a lot of work (including the literature cited by the authors) reporting ferroelectric topological domains with various microstructures, the solitons proposed by the authors. In this work, STEM and PFM results show there are several localized solitons in the BFO-STO superlattice. However, Based on the existing data and the author's response, I still can't conclude that PFM and STEM correspond to the same object. These solitons are either aperiodic or universal, just local structures. The effects of these solitons on the macroscopic properties of ferroelectric thin films are not given. So, I don't think that just finding several local soliton structures can be published in such a high-impact journal.

In reply 1, the situation of the literature listed by the author seems to be different from that of the author. The former is a periodic structure, or a topological structure in a single-layer PTO, and there is no superposition of multiple "solitons" in the thickness direction.

In reply 2, the author claims that it is the amplification of electric field in PFM. How does "zoo of structures" shown in PFM correspond to "zoo of structures" shown in planar and sectional STEM? It says create, disappear, merge, structural phase transition? In Fig.S5, it shouldn't be whether the cluster as a whole presents (skyrmions, meron, bimeron, or disclination)?

In reply 6, it is suggested that the author mark the interface in the HAADF images. In Fig S3, obvious interface fluctuation and interface diffusion exist, and the author claims that the interface sharp is inappropriate.

In reply 7, the topography of Fig. R9 shows that local surface topography exceeds 3uc , which is apparent compared to the BFO layer of 7uc thickness or the STO layer of 4uc thickness. This is not a high quality layer-by-layer growth, and this domain structures in PFM images is probably just a generic nanodomain in top-layer BFO of the BFO-STO superlattice.

Reviewer #2 (Remarks to the Author):

The authors have addressed my concerning points. I would like to recommend its publication in Nature Communications.

Reviewer #3 (Remarks to the Author):

The authors have well resolved my concerns in the revised manuscript. I suggest the manuscript be accepted after resolving other reviewers' concerns.

Reviewer #1 (Remarks to the Author):

As far as I know, there has been a lot of work (including the literature cited by the authors) reporting ferroelectric topological domains with various microstructures, the solitons proposed by the authors.

Response: We agree that currently the field of ferroelectric topologies is attracting a lot of attention. There have been many important breakthroughs; however, not one of them reports the types of ferroelectric topologies in a BFO-based system as we do. In Table RL1 below, we have listed the main papers published by leading research groups in chronological order. We have also included the material system studied, along with their key finding. This table demonstrates that not one of them presents findings similar to ours.

Table RL1. List of representative works of ferroelectric topological domains (sorted by publication year).

Title	Journal & published year	Corresponding authors	Materials Type	System	Topological domain type	Domain structure
1. Unusual phase transitions in ferroelectric nanodisks and nanorods.	Nature, 2004	L. Bellaiche, H. X. Fu	Ferroelectrics	Pb(Zr,Ti)O ₃ disks and Rods	Polar vortices	[Redacted]
2. Spontaneous Vortex Nanodomain Arrays at Ferroelectric Heterointerfaces	Nanoletters, 2011	Xiaoqing Pan	Multiferroics	BiFeO ₃ /TbScO ₃	Polar vortices	[Redacted]
3. Observation of a periodic array of flux-closure quadrants in strained ferroelectric PbTiO ₃ films.	Science, 2015	X. L. Ma, S. J. Pennycook	Ferroelectrics	PbTiO ₃ /SrTiO ₃ multilayer	Flux-closure domains	[Redacted]
4. Observation of polar vortices in oxide superlattices.	Nature, 2016	L. W. Martin, R. Ramesh	Ferroelectrics	PbTiO ₃ /SrTiO ₃ superlattice	Polar vortices	[Redacted]

5. Nanoscale bubble domains and topological transitions in ultrathin ferroelectric films.	Adv. Mater., 2017	L. Bellaiche, N. Valanoor	Ferroelectrics	Pb(Zr,Ti)O₃/SrTiO₃ multilayer	Polar bubbles	[Redacted]
6. High-density array of ferroelectric nanodots with robust and reversibly switchable topological domain states.	Sci. Adv., 2017	X. S. Gao, J. M. Liu	Multiferroics	BiFeO₃ nanodots	Center domains	[Redacted]
7. Rewritable ferroelectric vortex pairs in BiFeO₃	Npj Quantum Materials, 2017	Xiaomei Lu	Multiferroics	BiFeO₃	Vortex-antivortex pairs	8. Rhombohedral–orthorhombic ferroelectric morphotropic phase boundary associated with a polar vortex in BiFeO₃ Films.	ACS Nano, 2018	Y. L. Zhu, X. L. Ma	Multiferroics	BiFeO₃/GdScO₃ multilayer	Polar vortices	[Redacted]

9. Topological defects with distinct dipole configurations in PbTiO ₃ /SrTiO ₃ multilayer films.	Phys. Rev. Lett., 2018	C. L. Jia, L. Bellaiche	Ferroelectrics	PbTiO ₃ /SrTiO ₃ multilayer	Polar disclinations	[Redacted]
10. Defect-Induced Hedgehog Polarization States in Multiferroics.	Phys. Rev. Lett., 2018	X. Pan	Multiferroics	BiFeO ₃ films	Hedgehog domains	[Redacted]
11. Observation of room-temperature polar skyrmions.	Nature, 2019	L. W. Martin, R. Ramesh	Ferroelectrics	PbTiO ₃ /SrTiO ₃ superlattice	Polar skyrmions	[Redacted]
12. Optical creation of a supercrystal with three-dimensional nanoscale periodicity.	Nat. Mater., 2019	V. Gopalan, J. W. Freeland	Ferroelectrics	PbTiO ₃ /SrTiO ₃ superlattice	Polar supercrystal	[Redacted]
13. Polar meron lattice in strained oxide ferroelectrics.	Nat. Mater., 2020	Y. L. Zhu, X. L. Ma	Ferroelectrics	PbTiO ₃ films	Polar merons	[Redacted]

14. Hopfions emerge in ferroelectrics.	Nat. Commun., 2020	V. M. Vinokur	Ferroelectrics	Pb(Zr,Ti)O₃ nanoparticle	Polar hopfions	15. Inverse transition of labyrinthine domain patterns in ferroelectric thin films.	Nature, 2020	Y. Nahas, L. Bellaiche	Ferroelectrics /Multiferroics	PbTiO₃ and BiFeO₃ films	Labyrinthine domains	[Redacted]
16. Atomic mapping of periodic dipole waves in ferroelectric oxide.	Sci. Adv., 2021	Y. L. Zhu,X. L. Ma	Ferroelectrics	PbTiO₃/SrTiO₃ superlattice	Polar waves	[Redacted]

17. Creating polar antivortex in PbTiO₃/SrTiO₃ superlattice	Nat. Commun., 2021	J. Y. Li, P. Gao	Ferroelectrics	PbTiO₃/SrTiO₃ superlattice	Polar antivortex	18. Toroidal polar topology in strained ferroelectric polymer.	Science, 2021	C. W. Nan, Y. Shen	Ferroelectrics	P(VDF-TrFE)	Polar toroidal domains	[Redacted]
19. Chiral polarization textures induced by the flexoelectric effect in ferroelectric nanocylinders.	Phys. Rev. B, 2021	A. N. Morozovska	Ferroelectrics	BaTiO₃ nanoparticle	Polar flexon	[Redacted]

20. Vortex Domain Walls in Ferroelectrics.	Nano Lett., 2021	Z. Hong, R. Ramesh	Ferroelectrics	PbTiO₃/SrTiO₃ superlattice	Polar vortex domain walls	[Redacted]
21. Ferroelectric incommensurate spin crystals.	Nature, 2022	M. Alexe	Ferroelectrics	PbTiO₃/SrRuO₃ multilayer	Polar spin ice	[Redacted]

In this work, STEM and PFM results show there are several localized solitons in the BFO-STO superlattice. However, Based on the existing data and the author's response, I still can't conclude that PFM and STEM correspond to the same object. These solitons are either aperiodic or universal, just local structures.

Response: We will show in our later response that these are not just local structures. We are unsure as to why the Referee would insist that the PFM and STEM must correspond to the exact same domains, as it would be impossible to use PFM to image domains of the order of few nm and then use STEM to image the same domains. We also disagree with the Referee's assertions – "These solitons are either aperiodic or universal, just local structures." We presume the Referee means "neither aperiodic nor universal"- These are certainly not just several localised solitons. We will present enough evidence to prove that we are not selectively showing data.

The effects of these solitons on the macroscopic properties of ferroelectric thin films are not given. So, I don't think that just finding several local soliton structures can be published in such a high-impact journal.

Response: We do not agree with the statement that our work is "just finding several local soliton structures." We remind the Referees that even in the case of the ground-breaking PTO findings, the first few reports focused on understanding the formation of such non-trivial topologies. In our work, we have given a full detailed analysis of our findings to a level not seen previously for bismuth ferrite-strontium titanate superlattices. It is crucial to demonstrate an understanding of how these structures form – and under precisely which conditions – to be able to make useful property claims. These questions are exactly what this paper addresses.

[Redacted]

[Redacted]

R1. In reply 1, the situation of the literature listed by the author seems to be different from that of the author. The former is a periodic structure, or a topological structure in a single-layer PTO, and there is no superposition of multiple "solitons" in the thickness direction.

Response: We are sorry, but we do not understand. It would seem unreasonable to expect the same structures in two very different systems (BFO and PTO), especially since they have completely different crystallographic ground states (rhombohedral and tetragonal respectively). We did explain this reasoning in detail in our previous response, which we have summarized again below.

It is critical to understand why our system does display such a wide array of different topological structures, while the PTO-STO system displays only vortices in ordered arrays.

We believe the explanation lies in the wide array of strain and crystallographic structures that are possible in BFO. Depending on the imposed epitaxial strain, BFO can form in either R-like or T-like phases, and we observe evidence of both phases (albeit with a very tiny phase fraction of the T-like phase) from our XRD measurements (Fig. 1b). Recently we showed that through anisotropic strain, one can stabilize a new triclinic phase of BFO. This proves that the material is rather compliant and "malleable" for strain and crystallographic structures.

A further important distinction between the BFO and PTO systems is the fact that BFO is rhombohedral in bulk (i.e., with three cartesian components of the polarization) while PZT with large Ti composition or PTO is tetragonal in bulk (i.e., with only one cartesian component of the polarization). It is quite likely that this fact makes BFO richer for ferroelectric solitons simply because there are more possible orientations of the polarization vector.

Finally, the electric dipoles of BFO also compete or interact with oxygen octahedral tilting and magnetism, whereas this does not occur in PTO. We believe that it is also possible that this makes BFO richer for topological defects.

R2. In reply 2, the author claims that it is the amplification of electric field in PFM. How does "zoo of structures" shown in PFM correspond to "zoo of structures" shown in planar and sectional STEM?

Response: We thank the Referee for this very important question. To answer it, we show STEM from three different regions (Figs. RL2-RL4) and PFM (Fig. RL5) from three different regions of the same sample. We can see that first, the solitons are found everywhere and are not restricted to some selective layers or regions. In all cases across the sample we observe a variety of polar textures no matter how the polarization is imaged. We can find a zoo; viz. merons, skyrmions as well as centre-divergent and convergent structures, not separate but interconnected to each other through a complex dipolar network. These STEM images are consistent with the PFM observations: the PFM images show (i) a similar mixture of complex topologies but at a larger length scale hinting at topological protection and (ii) several types on nanodomains that are connected to each other through a complex arrangement of disclinations and bimerons. In fact at the end of the results and discussion of the PFM section in our manuscript we had previously highlighted the occurrence of "transitional topological states" and "disclinations". This link has now been highlighted stronger in the "Atomic Scale Characterisation" section of our manuscript.

Changes made to manuscript:

We have added the following short discussion:

"Nevertheless the STEM images are consistent with the previous PFM data. Both techniques show a range of topologies of varying sizes irrespective of the imaging technique. Second, both find that the solitons are connected to each other through a complex dipolar network consisting of bimerons and disclination type defects."

Figure RL2. Planar view HAADF-STEM imaging of the BFO/STO superlattice (a) low magnification view. (b) Magnifications of areas I, II and III labelled in a, showing that the solitons are formed at all locations. (c) Magnifications of area (marked by a yellow box) in b and corresponding polar map, revealing the different solitons.

Figure RL3. Cross-section HAADF-STEM imaging of the BFO/STO superlattice (a) Low magnification view. (b) Magnifications of areas I, II and III labelled in a and corresponding polar maps, showing the different solitons (marked by black arrows).

Figure RL4. (a) Low-magnification cross-section view of HAADF-STEM image of the BFO/STO superlattice. Scale bar, 20 nm. (b) Magnifications of areas labelled in a, showing sharp interfaces between BFO and STO. Scale bar, 5 nm.

Changes made to manuscript:

Supplementary S3 and S7 have been modified to include the dashed lines as required by the Referee, and we have included the additional STEM data from different regions.

It says create, disappear, merge, structural phase transition? In Fig.S5, it shouldn't be whether the cluster as a whole presents (skyrmions, meron, bimeron, or disclination)?

Response: We do apologize but the meaning of the above comments is unclear to us.

In reply 6, it is suggested that the author mark the interface in the HAADF images. In Fig S3, obvious interface fluctuation and interface diffusion exist, and the author claims that the interface sharp is inappropriate.

Response: As requested we have marked the interface lines (see below Fig. S3b).

Fig. S3 (new) | Structural characterization of a $(\text{BFO}_7/\text{STO}_4)_{10}$ superlattice. (a) Cross-sectional high-angle annular dark-field (HAADF)-STEM image. Dashed lines denote the interfaces between BFO and STO. Scale bar: 5 nm. (b-c) EDS-mapping of BFO/STO superlattice. Scale bar: 2 nm. (b) Overlapped EDS mapping of Ti, Fe, Sr, and Bi elements. (c) Individual Ti, Fe, Sr, and Bi element maps in the same location as shown in (b).

We also show the interface in six regions as in Fig. RL4.

In reply 7, the topography of Fig. R9 shows that local surface topography exceeds $3uc$, which is apparent compared to the BFO layer of $7uc$ thickness or the STO layer of $4uc$ thickness. This is not a high quality layer-by-layer growth, and this domain structures in PFM images is probably just a generic nanodomain in top-layer BFO of the BFO-STO superlattice.

Response: We strongly disagree with this statement. Our data clearly show that this is not a generic nanodomain in the top-layer BFO of the superlattice. First, in response R2 we show sufficient STEM data from different regions (Figs. RL2 and RL3) that **these are domains are spread everywhere and found in several layers below the surface**. Second, in Fig. RL5 we show PFM from six completely different regions of the same sample. From these scans, we determine the average roughness of the sample surface (Fig. RL6) – here it is highly evident that the roughness is at most 0.22 nm (just over **half of one unit cell**), with the median value being only 0.16 nm (below **half of one unit cell**). We can therefore conclude that this sample has an **atomically smooth surface**.

Figure RL5. Piezoresponse force microscopy (PFM) amplitude and phase images of different regions in same BFO-STO superlattice sample. Panels (A) and (B) show low and high magnification scans of six different spots.

Figure RL6. Surface roughness values taken from six different regions (shown in Fig. RL5), as a box-and-whisker plot. Both the mean and median are around 0.16 nm, well below a single unit cell (0.4 nm).

Finally, we draw a comparison of the PFM of BFO superlattice sample in our paper (with solitons) to a sample where the BFO thickness in the superlattice layers has been increased (i.e., out of the stability window for formation of solitons), see Fig. RL7. There is a clear difference in the topologies observed.

In summary, the PFM images (Fig. RL5) taken in conjunction with the new STEM data from different regions (Figs. RL2 and RL3) unambiguously demonstrate that our observed topological solitons are universal for this sample.

No changes to the manuscript have been made.

Figure RL7. PFM scans for BFO/STO superlattice heterostructures with varying BFO layer thickness. [7/4]10 BFO/STO sample (top row) clearly shows smaller circular-like domains, whereas in [9/4]10 BFO/STO sample (bottom row), the coexistence of labyrinthine and circular-shaped domain becomes more apparent.